# Gradient-Driven Rewards to Guarantee Fairness in Collaborative Machine Learning

**Xinyi Xu**[1,5]*, **Lingjuan Lyu**[2]*, **Xingjun Ma**[3], **Chenglin Miao**[4],
**Chuan Sheng Foo**[5], **and Bryan Kian Hsiang Low**[1]

Department of Computer Science, National University of Singapore, Republic of Singapore[1]
Sony AI[2], School of Computer Science, Fudan University, People's Republic of China[3]
Department of Computer Science, University of Georgia, USA[4]
Institute for Infocomm Research, A*STAR, Republic of Singapore[5]
`{xuxinyi,lowkh}@comp.nus.edu.sg`[1], `lingjuan.lv@sony.com`[2]
`danxjma@gmail.com`[3], `cmiao@uga.edu`[4], `foo_chuan_sheng@i2r.a-star.edu.sg`[5]

## Abstract

In *collaborative machine learning* (CML), multiple agents pool their resources (e.g., data) together for a common learning task. In realistic CML settings where the agents are self-interested and not altruistic, they may be unwilling to share data or model information without adequate rewards. Furthermore, as the data/model information shared by the agents may differ in quality, designing rewards which are fair to them is important so that they would not feel exploited nor discouraged from sharing. In this paper, we adopt *federated learning* as the CML paradigm, propose a novel *cosine gradient Shapley value* (CGSV) to fairly evaluate the expected marginal contribution of each agent's uploaded model parameter update/gradient without needing an auxiliary validation dataset, and based on the CGSV, design a novel training-time gradient reward mechanism with a fairness guarantee by sparsifying the aggregated parameter update/gradient downloaded from the server as reward to each agent such that its resulting quality is commensurate to that of the agent's uploaded parameter update/gradient. We empirically demonstrate the effectiveness of our fair gradient reward mechanism on multiple benchmark datasets in terms of fairness, predictive performance, and time overhead.

## 1 Introduction

In *collaborative machine learning* (CML), multiple agents (e.g., researchers, organizations, companies) pool their resources (e.g., data) together for a common learning task. It spans a wide variety of real-world applications such as digital healthcare [49], clinical trial research [13, 23], wake word detection for smart voice assistants [27], and next word prediction on mobile devices [15].

*Federated learning* (FL) provides a natural paradigm of CML [18, 29, 41, 43, 57, 62]. In FL, the agents perform local model training (e.g., using stochastic gradient descent) and share their resulting model parameter updates/gradients via a *trusted server* [40, 56, 59]. An important distinction of our work here from the standard FL literature is that the agents are self-interested and hence not necessarily cooperative like the worker nodes in distributed learning. The implication is that to achieve competitive predictive performance for the learning task, it is imperative to incentivize/reward the agents for contributing/sharing high-quality information in the form of model parameter updates/gradients [47, 48, 52].

---

*Equal contribution.

35th Conference on Neural Information Processing Systems (NeurIPS 2021).

Our work here adopts FL as the CML paradigm for designing a fair reward mechanism such that the (self-interested) agents who contribute more would not feel exploited but be rewarded commensurately. This is often regarded as fairness in cooperative game theory [42], mechanism design [4], and computational social choice [11]. To design such a fair reward mechanism, we need to address three main questions:

Firstly, *what is a suitable notion of fairness?* The *Shapley value* (SV) [50] from cooperative game theory is an appealing choice and has been used in ML [14] and FL [54, 56]. However, existing SV-based works [19, 37, 54, 56] typically require the availability of (and all agents to agree on) an auxiliary validation dataset and significant time overhead from evaluating the agents' contributions in the form of SVs and the resulting model training. To overcome these difficulties, we propose to instead exploit the alignment (specifically, cosine similarity) of an agent's uploaded/contributed model parameter update/gradient vector (or that aggregated over some agents) to that aggregated over all agents (hence measuring its quality/value and circumventing the need for a validation dataset [12, 52]) for devising our proposed *cosine gradient Shapley value* (CGSV) (Sec. 3.2) which can be efficiently approximated with a bounded error (Sec. 3.3).

Secondly, *what is the choice of reward?* Various choices such as monetary rewards from a pre-allocated budget [65, 66] or the total revenue generated from the collaboration through FL [9, 10] have been proposed. Though it may seem natural to consider monetary rewards, it is not obvious how a common denomination between money and data/gradients [1, 46] can be readily established, which makes it challenging to apply these works in practice. Instead, we propose to consider the aggregated parameter updates/gradients downloaded from the server as rewards to the agents.

Finally, *how can the gradient reward mechanism ensure fairness?* Our proposed mechanism exploits a *sparsifying gradient* trick (Sec. 3.4) for controlling the quality of the aggregated parameter update/gradient downloaded from the server as reward to each agent at training time (rather than post hoc [48, 52, 65]) such that its quality is commensurate to that of the agent's uploaded/contributed parameter update/gradient [2, 7]. Consequently, an agent who uploads/contributes higher-quality parameter updates/gradients over the entire training process should eventually be rewarded with converged model parameters whose resulting training loss (and hence predictive performance) is closer to that of the server, as demonstrated in our fairness guarantee (Sec. 3.5) [52].

In summary, the contributions of our work here to CML and FL include the following:

- We propose a novel *cosine gradient Shapley value* (CGSV) (Sec. 3.2) to fairly evaluate the expected marginal contribution of each agent's uploaded model parameter update/gradient without needing an auxiliary validation dataset and present an efficient approximation of CGSV with a bounded error (Sec. 3.3).
- Based on the approximate CGSV, we design a novel training-time gradient reward mechanism (Sec. 3.4) with a fairness guarantee (Sec. 3.5) by exploiting the trick of sparsifying the aggregated parameter update/gradient downloaded from the server as reward to each agent such that its resulting quality is commensurate to that of the agent's uploaded/contributed parameter update/gradient.
- We empirically demonstrate the effectiveness of our fair gradient reward mechanism on multiple benchmark datasets in terms of fairness, predictive performance, and time overhead (Sec. 4).

## 2   Related Work

**Reward design and choice in CML.** In related topics such as FL [30, 36, 38, 47, 59, 63, 66], Bayesian CML [52], collaborative generative modeling [55], and data sharing [13, 23, 48], designing appropriate rewards to encourage collaboration (e.g., sharing real or synthetic data, gradients, or other information) is a non-trivial problem. A useful solution concept should provide a formal notion of fairness, a suitable form/denomination of reward, and a principled way to guarantee fairness via a carefully designed reward mechanism. Previous works have considered monetary rewards from a pre-allocated budget [65, 66] or the total revenue generated from the collaboration [9, 10], or simply an abstract yet quantifiable form of reward [47, 48]. Though it may seem natural to consider monetary rewards, it is not obvious how a common denomination between money and data/gradients [1, 46] can be readily established, which makes it challenging to apply these works in practice. The work of [66] has explored a different avenue of using a reverse auction to guarantee truthfulness in its mechanism instead of fairness.

**Fairness notions.** The *Shapley value* (SV) [50] from cooperative game theory is widely regarded as a principled notion of fairness [4, 11, 42] due to its several desirable properties such as symmetry and null player. Existing SV-based works have considered fairness in the sense of rewarding agents according to their contributions [19, 54, 56]. However, they typically require the availability of (and all agents to agree on) an auxiliary validation dataset [37, 52] and significant time overhead from evaluating the agents' contributions in the form of SVs and the resulting model training [14, 19, 56]. In contrast, the work of [31] has adopted an egalitarian notion of fairness by aiming to equalize the final individual performance among agents, which is fundamentally different from SV.

Different from the fairness definition in [31], we adopt a fairness notion formalized by SV [14, 19, 52, 54, 56]. Our proposed work is novel in the application of SV: While previous works use the validation accuracy [14, 19, 54, 56], we exploit the cosine similarity between model parameter updates/gradient vectors [12] for devising our proposed *cosine gradient Shapley value* (CGSV) (Sec. 3.2) to fairly evaluate the expected marginal contribution of each agent's uploaded model parameter update/gradient. Based on the CGSV, we design a novel training-time gradient reward mechanism (Sec. 3.4) with a fairness guarantee (Sec. 3.5) and empirically show that it outperforms several existing FL baselines in terms of predictive performance, fairness, and time overhead (Sec. 4.2).

## 3 Fair Gradient Reward Mechanism

### 3.1 Vanilla Federated Learning (FL) Problem Setting and Notations

The vanilla FL problem [56, 59] involves a set $\mathcal{N} := \{i\}_{i=1,\ldots,N}$ of $N$ *honest* agents learning a $D$-dimensional vector $\boldsymbol{w} \in \mathbb{R}^D$ of model parameters to minimize a loss function $\mathbf{F}(\boldsymbol{w})$ that can be additively decomposed into $N$ local differentiable loss functions $\mathbf{F}_i(\boldsymbol{w})$ defined using the local dataset $\mathcal{D}_i$ of agent $i \in \mathcal{N}$ and weighted by its importance $p_i \geq 0$ (e.g., proportional to $|\mathcal{D}_i|$). That is, $\mathbf{F}(\boldsymbol{w}) := \sum_{i \in \mathcal{N}} p_i \, \mathbf{F}_i(\boldsymbol{w})$ where $\sum_{i \in \mathcal{N}} p_i = 1$. We call $\mathcal{N}$ the grand coalition; a coalition $\mathcal{S} \subseteq \mathcal{N}$ is then a subset of the grand coalition $\mathcal{N}$ of $N$ agents. In iteration $t = 0$, every agent $i \in \mathcal{N}$ starts with the same initialized parameter vector $\boldsymbol{w}_{i,0} := \boldsymbol{w}_0$ as the server. In iteration $t > 0$, every agent $i \in \mathcal{N}$ calculates a parameter update $\Delta \boldsymbol{w}_{i,t} := -\eta_t \nabla \mathbf{F}_i(\boldsymbol{w}_{i,t-1})$ with step size $\eta_t$ and gradient $\nabla \mathbf{F}_i(\boldsymbol{w}_{i,t-1})$ w.r.t. parameter vector $\boldsymbol{w}_{i,t-1}$ and uploads it to a *trusted* server who normalizes and aggregates all agents' parameter updates as follows:

$$\boldsymbol{u}_{i,t} := \Gamma \, \Delta \boldsymbol{w}_{i,t} / \|\Delta \boldsymbol{w}_{i,t}\|, \quad \boldsymbol{u}_{\mathcal{N},t} := \sum_{i \in \mathcal{N}} r_{i,t-1} \, \boldsymbol{u}_{i,t} \tag{1}$$

where $\Gamma$ is a normalization coefficient used to prevent gradient explosion [33, 45] and the importance coefficient $r_{i,t-1}$ will be described later in Sec 3.4. So, we call (1) the *gradient aggregation step*. The *gradient download step* then follows where every agent $i \in \mathcal{N}$ downloads the aggregated parameter update/gradient $\boldsymbol{u}_{\mathcal{N},t}$ (1) from the server (as reward) for updating its model parameters $\boldsymbol{w}_{i,t} := \boldsymbol{w}_{i,t-1} + \boldsymbol{u}_{\mathcal{N},t}$ to the same $\boldsymbol{w}_t := \boldsymbol{w}_{t-1} + \boldsymbol{u}_{\mathcal{N},t}$ as the server. That is, $\boldsymbol{w}_{i,t} = \boldsymbol{w}_t$ for all $i \in \mathcal{N}$ and $t \in \mathbb{Z}^+ \cup \{0\}$. We define $\boldsymbol{u}_{\mathcal{S},t}$ for any coalition $\mathcal{S} \subseteq \mathcal{N}$ in a similar way as $\boldsymbol{u}_{\mathcal{N},t}$ (1). For brevity, we omit $t$ from our notations in Secs. 3.2 and 3.3 since we only refer to iteration $t$.

### 3.2 Cosine Gradient Shapley Value (CGSV) for Fairness

In the gradient aggregation step (1), the quality/value of coalition $\mathcal{S}$'s (normalized) aggregated parameter update/gradient $\boldsymbol{u}_{\mathcal{S}}$ can be measured by its *cosine similarity* $\cos(\boldsymbol{u}_{\mathcal{S}}, \boldsymbol{u}_{\mathcal{N}}) := \langle \boldsymbol{u}_{\mathcal{S}}, \boldsymbol{u}_{\mathcal{N}} \rangle / (\|\boldsymbol{u}_{\mathcal{S}}\| \|\boldsymbol{u}_{\mathcal{N}}\|)$ to the grand coalition $\mathcal{N}$'s aggregated parameter update/gradient $\boldsymbol{u}_{\mathcal{N}}$ [12, 28, 35]. We use this cosine similarity measure as our *gradient valuation function* $\nu(\mathcal{S}) := \cos(\boldsymbol{u}_{\mathcal{S}}, \boldsymbol{u}_{\mathcal{N}})$. Intuitively, if the direction of $\boldsymbol{u}_{\mathcal{S}}$ aligns more closely with that of $\boldsymbol{u}_{\mathcal{N}}$, then its quality/value $\nu(\mathcal{S})$ is higher. Using $\nu$, the contribution $\phi_i$ of agent $i \in \mathcal{N}$ is defined based on the notion of *Shapley value* (SV) [50] which measures its expected marginal contribution when joining the other agents preceding it in any permutation and satisfies certain desirable fairness properties [5], such as null player (i.e., an agent with no marginal contribution has zero SV), symmetry (i.e., agents with identical marginal contributions have equal SVs), among others, as formally discussed in Appendix A.1:

**Definition 1** (**Cosine gradient Shapley value (CGSV)**). Let $\Pi_{\mathcal{N}}$ be a set of all possible permutations of $\mathcal{N}$ and $\mathcal{S}_{\pi,i}$ be the coalition of agents preceding agent $i$ in permutation $\pi \in \Pi_{\mathcal{N}}$. The CGSV of agent $i \in \mathcal{N}$ is defined as

$$\phi_i := (1/N!) \sum_{\pi \in \Pi_{\mathcal{N}}} \left[ \nu(\mathcal{S}_{\pi,i} \cup \{i\}) - \nu(\mathcal{S}_{\pi,i}) \right]. \tag{2}$$

If $\phi_i$ is negative, then it follows from the weighted sum of parameter updates/gradients in (1) that $\boldsymbol{u}_i$ points in an opposite direction to some other parameter updates/gradients and hence has negative cosine similarities to them. In practice, due to the noisy training arising from the use of *stochastic gradient descent* (SGD) and/or a highly non-convex loss function, $\phi_i$ may at times be negative even for an honest agent $i$. When the number of such cases is limited, training via SGD can still converge to yield a competitive predictive performance, as empirically validated in [12].

### 3.3 Efficient Approximation of CGSV

Since evaluating agent $i$'s CGSV $\phi_i$ (2) exactly incurs $\mathcal{O}(2^N D)$ time and is thus costly, we propose an efficient approximation by directly measuring the cosine similarity of its (normalized) parameter update/gradient $\boldsymbol{u}_i$ to the grand coalition $\mathcal{N}$'s aggregated parameter update/gradient $\boldsymbol{u}_\mathcal{N}$, which reduces the incurred time by a factor of $2^N$ and has a bounded error from $\phi_i$ (Theorem 1):

$$\phi_i \approx \psi_i \coloneqq \cos(\boldsymbol{u}_i, \boldsymbol{u}_\mathcal{N}) . \tag{3}$$

**Theorem 1 (Approximation Error).** *Let $I \in \mathbb{R}^+$. Suppose that $\|\boldsymbol{u}_i\| = \Gamma$ and $|\langle \boldsymbol{u}_i, \boldsymbol{u}_\mathcal{N}\rangle| \geq 1/I$ for all $i \in \mathcal{N}$. Then, $\phi_i - L_i\psi_i \leq I\Gamma^2$ where the multiplicative factor $L_i$ can be normalized away.*

Its proof is in Appendix A.2. From Theorem 1, the approximation error is bounded and decreases quadratically with normalization coefficient $\Gamma$. However, $\Gamma$ cannot be reduced to be arbitrarily small, which may cause $|\langle \boldsymbol{u}_i, \boldsymbol{u}_\mathcal{N}\rangle| \geq 1/I$ not to hold. It also does not hold when $\boldsymbol{u}_i$ is orthogonal to $\boldsymbol{u}_\mathcal{N}$ or is close to the zero vector, hence implying the quality of that agent $i$'s parameter update/gradient is not high enough. So, every agent is encouraged to contribute a parameter update/gradient of sufficiently high quality in order to ensure the quality of the approximation $\psi_i$ (Theorem 1).

We have performed a simple experiment to compare the quality of our approximation $\psi_i$ with that of a sampling-based $(\epsilon, \delta)$-approximation $\bar{\phi}_i$ [39], the latter of which is widely used by existing works in data valuation and CML/FL [14, 19, 52, 56, 60]. In this experiment, we have drawn $N$ random $D$-dimensional vectors from a standard multivariate normal distribution to simulate $\boldsymbol{u}_1, \ldots, \boldsymbol{u}_N$ and calculated the resulting exact CGSVs $\boldsymbol{\phi} \coloneqq (\phi_i)_{i=1,\ldots,N}$, our approximation $\boldsymbol{\psi} \coloneqq (\psi_i)_{i=1,\ldots,N}$, and the sampling-based $(0.1, 0.1)$-approximation $\bar{\boldsymbol{\phi}} \coloneqq (\bar{\phi}_i)_{i=1,\ldots,N}$. Fig. 1 shows the results for $\ell_1$ error, $\ell_2$ error, and the incurred time averaged over 10 runs: Our approximation $\boldsymbol{\psi}$ performs better in all three metrics with varying $D$ (right figure) and the performance gap widens with an increasing number $N$ of agents (left figure).

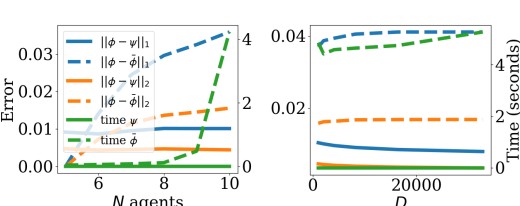

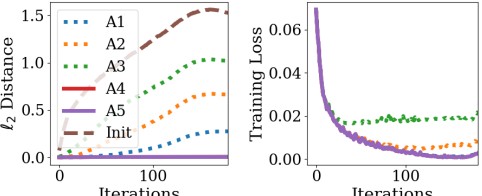

Figure 1: Comparison of $\ell_1$ error (blue), $\ell_2$ error (orange), and incurred time (green) (i.e., averaged over 10 runs) between our approximation $\boldsymbol{\psi}$ (solid lines) vs. a sampling-based approximation $\bar{\boldsymbol{\phi}}$ (dashed lines) [39] of the exact CGSVs $\boldsymbol{\phi}$ with (left) varying number $N$ of agents and $D = 1024$, and (right) varying vector dimension $D$ and $N = 10$. For all metrics, lower is better.

Figure 2: (Left) $\ell_2$ distance between model parameters of agent $i = 1, \ldots, 5$ (abbreviated to A$i$) vs. that of the server, and (right) corresponding training loss for an FL problem with $N = 5$ agents using local MNIST datasets of 600 images each to collaboratively learn 2-layer CNN parameters where the datasets of A1 (blue), A2 (orange), and A3 (green) have $20\%$, $40\%$, and $60\%$ randomly corrupted labels, respectively. The brown line denotes $\ell_2$ distance between $\boldsymbol{w}_0$ (initialization) vs. server's model parameters.

### 3.4 Server-Side Training-Time Gradient Reward Mechanism

We will now describe the exact details of the gradient aggregation and download steps performed by the server to implement our proposed fair gradient reward mechanism:

**Gradient Aggregation Step.** With a specified normalization coefficient $\Gamma$ and an initialized coefficient $r_{i,0}$, the server performs normalization and aggregation of all agents' parameter updates into $\boldsymbol{u}_{\mathcal{N},t}$ using (1), as previously discussed in the FL problem setting (Sec. 3.1). Then, the server computes our approximation $\psi_{i,t}$ (3) of the CGSV $\phi_{i,t}$ (2) and updates (and normalizes) the importance coefficient $r_{i,t}$ in iteration $t$ via a moving average of $\psi_{i,t}$ given the relative weight $\alpha$ on $r_{i,t-1}$ from previous iteration $t-1$:

$$r_{i,t} := \alpha \, r_{i,t-1} + (1-\alpha) \, \psi_{i,t} \,, \quad r_{i,t} \leftarrow r_{i,t} / \textstyle\sum_{i' \in \mathcal{N}} r_{i',t} \qquad (4)$$

where $r_{i,0} := 0$. Note that $r_{i,t}$ (4) is used for deriving the sparsified gradient (5) in the gradient download step as well as the aggregation of all agents' parameter updates into $\boldsymbol{u}_{\mathcal{N},t+1}$ (1) in iteration $t+1$. The use of a moving average of $\psi_{i,t}$ to compute $r_{i,t}$ (4) provides a smoothed estimate without abrupt fluctuations and reduces the effect of noisy training due to the use of SGD in practice [31, 56]. It also allows a flexible weighting over the iterations of the entire training process: In particular, setting $\alpha < 1$ can effectively mitigate the noise from random initialization of model parameters $\boldsymbol{w}_0$ because the weight on $\psi_{i,t'}$ in earlier iteration $t' < t$ decays exponentially with $t$ [54].

**Gradient Download Step.** Recall from the *vanilla* FL problem setting (Sec. 3.1) that in each iteration $t$, this step involves all agents downloading an identical aggregated parameter update/gradient $\boldsymbol{u}_{\mathcal{N},t}$ (1) from the server (as reward) for updating their model parameters to the same $\boldsymbol{w}_t$ (as the server), which is expected to converge to yield a competitive predictive performance [8, 32]. However, such *equal* rewards to all agents is unfair and will discourage any agent from uploading/contributing a parameter update/gradient of higher quality [37, 63] when it can afford to. To ensure fairness, each agent should download some form of aggregated parameter update/gradient as reward that is commensurate to the quality/value of its uploaded/contributed parameter update/gradient. Consequently, an agent who uploads/contributes higher-quality parameter updates/gradients over the entire training process should eventually be rewarded with converged model parameters whose resulting training loss (and hence predictive performance) is closer to that of the server (Theorem 2).

To achieve this, we adopt the trick of *sparsifying*[2] the aggregated parameter update/gradient $\boldsymbol{u}_{\mathcal{N},t}$ downloaded from the server as reward to agent $i$ in each iteration $t$. Specifically, we zero out fewer of its smallest components (hence higher-quality gradient reward) when the importance coefficient $r_{i,t}$ (4) (i.e., moving average of the approximate CGSV $\psi_{i,t}$) is larger:

$$\boldsymbol{v}_{i,t} := \text{mask}(\boldsymbol{u}_{\mathcal{N},t}, q_{i,t}) \,, \quad q_{i,t} := \lfloor D \tanh(\beta \, r_{i,t}) / \max_{i' \in \mathcal{N}} \tanh(\beta \, r_{i',t}) \rfloor \qquad (5)$$

where $\text{mask}(\boldsymbol{u}, q)$ retains the largest $\max(0, q)$ components (in magnitude) of $\boldsymbol{u}$ and zeros out all of its other components [2, 61], and $\beta \geq 1$ specifies the degree of altruism: Greater altruism $\beta$ gives any agent with a smaller $r_{i,t}$ a larger improvement in the quality of its gradient reward, i.e., a larger reduction in the sparsity of its downloaded $\boldsymbol{v}_{i,t}$ as reward. In the extreme case of $\beta = \infty$, we recover the *vanilla* FL problem setting (Sec. 3.1) where all agents are rewarded equally with $\boldsymbol{u}_{\mathcal{N},t}$ (i.e., best-quality gradient reward $\boldsymbol{v}_{i,t} = \boldsymbol{u}_{\mathcal{N},t}$ for all $i \in \mathcal{N}$ with no sparsification), albeit with importance coefficients $r_{i,t}$ possibly differing across agents $i \in \mathcal{N}$ and dynamically updated over iteration $t \in \mathbb{Z}^+$. Hence, increasing $\beta$ from 1 to $\infty$ trades off fairness for equality in gradient rewards by being more altruistic to any agent with a smaller $r_{i,t}$; we empirically show the effect of varying $\beta$ on training loss in Fig. 7 of Sec. 4.2. Note the agent $i^* := \text{argmax}_{i' \in \mathcal{N}} \tanh(\beta \, r_{i',t})$ with the largest possible $r_{i^*,t}$ does not benefit from such altruism since it already downloads the best-quality gradient reward (i.e., $\boldsymbol{u}_{\mathcal{N},t}$) according to (5).

Suppose that there exists a known threshold $\underline{r} > 0$ s.t. $r_{i,t} \geq \underline{r}$ for all $i \in \mathcal{N}$ and $t \in \mathbb{Z}^+$ and we want to limit the sparsity of any downloaded $\boldsymbol{v}_{i,t}$ or, equivalently, ensure the minimum quality of any gradient reward: Specifically, given a predefined threshold $c \in (0, 1]$, we want to guarantee $q_{i,t} \geq \lfloor D \times c \rfloor$ holds for all $i \in \mathcal{N}$ and $t \in \mathbb{Z}^+$. By setting $\beta$ s.t. $\tanh(\beta \, \underline{r}) \geq c$, it follows from (5) and $\max_{i' \in \mathcal{N}} \tanh(\beta \, r_{i',t}) \leq 1$ that $\tanh(\beta \, r_{i,t}) / \max_{i' \in \mathcal{N}} \tanh(\beta \, r_{i',t}) \geq \tanh(\beta \, r_{i,t}) \geq \tanh(\beta \, \underline{r}) \geq c$ and hence $q_{i,t} \geq \lfloor D \times c \rfloor$ ensues. By using the property that $\tanh(\beta \, \underline{r}) = (\exp(2\beta \, \underline{r}) - 1) / (\exp(2\beta \, \underline{r}) + 1)$, $\beta \geq \ln((1 + c)/(1 - c))/(2\underline{r})$ can be derived and used for setting $\beta$. It further informs us that reducing the sparsity of any downloaded $\boldsymbol{v}_{i,t}$ or, equivalently, improving the minimum quality of any gradient reward (i.e., by increasing $c$) requires greater altruism $\beta$ to be introduced, while improving the minimum quality of uploaded/contributed parameter updates/gradients by any agent over the entire training process (hence larger $\underline{r}$) eases the need of introducing greater altruism $\beta$.

---

[2]Sparsifying a parameter update/gradient vector means zeroing out some of its components and leaving the others unchanged [7, 33].

To see why the sparsifying gradient trick (5) can ensure fairness, we illustrate its effect in an FL problem with $N = 5$ agents using local MNIST datasets of 600 images each to collaboratively learn the parameters of a 2-layer *convolutional neural network* (CNN) where the datasets of agents 1, 2, and 3 have 20%, 40%, and 60% randomly corrupted labels, respectively. The uploaded/contributed parameter updates/gradients thus decrease in quality from agents 1 to 3 (i.e., $\psi_{1,t} = 0.194$, $\psi_{2,t} = 0.088$, and $\psi_{3,t} = 0.043$ on average) due to increasingly noisy labels in their datasets, while agents 4 and 5 upload/contribute parameter updates/gradients of high quality (i.e., $\psi_{4,t} = 0.331$ and $\psi_{5,t} = 0.342$ on average) due to uncorrupted labels in their datasets. Consequently, agents 1 to 3 have increasing sparsity (resp., 34.9%, 67.6%, and 83.0% on average) while agents 4 and 5 have little/no sparsity (resp., 3.5% and 1.1% on average) in their downloaded $\boldsymbol{v}_{i,t}$ as rewards ($\beta = 1$). Fig. 2 shows that the converged model parameters of agents 1 to 3 grow in $\ell_2$ distance from that of the server (hence increasing training loss) while agents 4 and 5 have the closest converged model parameters (hence lowest training loss).

We provide the pseudocodes performed by the server and agent $i \in \mathcal{N}$ in each iteration $t$ below. We will discuss in Sec 4.2 how the hyperparameters $\Gamma$ in (1), $\alpha$ in (4), and $\beta$ in (5) are set in our experiments.

---

**Server $(t)$**

---

1: **for all** $i \in \mathcal{N}$ **do**
2:     Download $\Delta \boldsymbol{w}_{i,t}$ from agent $i$
3:   ▷ **Gradient Aggregation Step**
4: Compute $\boldsymbol{u}_{i,t}$ and $\boldsymbol{u}_{\mathcal{N},t}$ (1)
5: **for all** $i \in \mathcal{N}$ **do**
6:     Compute $\psi_{i,t}$ (3) and $r_{i,t}$ (4)
7:   ▷ **Gradient Download Step**
8: **for all** $i \in \mathcal{N}$ **do**
9:     Compute $\boldsymbol{v}_{i,t}$ (5) for download by agent $i$

---

**Agent $(i, t)$**

---

1: Upload $\Delta \boldsymbol{w}_{i,t} = -\eta_t \nabla \mathbf{F}_i(\boldsymbol{w}_{i,t-1})$ to server
2: Download $\boldsymbol{v}_{i,t}$ from server
3: Update $\boldsymbol{w}_{i,t} = \boldsymbol{w}_{i,t-1} + \boldsymbol{v}_{i,t}$

---

### 3.5 Fairness Guarantee

We have previously discussed the intuition underlying our notion of fairness in Sec. 3.4 that an agent who uploads/contributes higher-quality parameter updates/gradients over the entire training process should eventually be rewarded with converged model parameters whose resulting training loss (and hence predictive performance) is closer to that of the server. Note that the importance coefficient $r_{i,t}$ (4) measures the overall quality of the parameter updates/gradients uploaded/contributed by agent $i$ over the entire training process till iteration $t$. Our main result below guarantees a notion of fairness that under some conditions on loss function $\mathbf{F}$ and the server's model parameters $\boldsymbol{w}_t$, if an agent $i$ has a larger importance coefficient $r_{i,t}$ and model parameters $\boldsymbol{w}_{i,t-1}$ closer to that of the server (i.e., $\boldsymbol{w}_{t-1}$) than another agent by at least $2\|\boldsymbol{v}_{i,t}\|$ in previous iteration $t-1$, then it is rewarded with model parameters $\boldsymbol{w}_{i,t}$ incurring smaller training loss $\mathbf{F}(\boldsymbol{w}_{i,t})$ in iteration $t$:

**Theorem 2 (Fairness in Training Loss).** *Let $\delta_{i,t} := \|\boldsymbol{w}_t - \boldsymbol{w}_{i,t}\|$. Suppose that $\boldsymbol{w}_t$ is near to a stationary point of $\mathbf{F}$ for $t \geq t^* \in \mathbb{Z}^+$ and some regularity conditions on $\mathbf{F}$ hold. For all $i, i' \in \mathcal{N}$ and $t \geq t^*$, if $r_{i,t} \geq r_{i',t}$ and $\delta_{i',t-1} - \delta_{i,t-1} \geq 2\|\boldsymbol{v}_{i,t}\|$, then $\mathbf{F}(\boldsymbol{w}_{i,t}) \leq \mathbf{F}(\boldsymbol{w}_{i',t})$.*

Its proof is in Appendix A.3. Our experiments in Appendix B.3 will empirically verify the fairness guarantee in Theorem 2 (and fairness in test accuracy) without needing to impose its conditions.

## 4 Experiments and Discussion

### 4.1 Experimental Settings

**Datasets.** We perform extensive experiments on image classification datasets like MNIST [26] and CIFAR-10 [21] and text classification datasets like *movie review* (MR) [44] and *Stanford sentiment treebank* (SST) [20]. We use a 2-layer *convolutional neural network* (CNN) for MNIST [25], a 3-layer CNN for CIFAR-10 [22], and a text embedding CNN for MR and SST [20].

**Baselines.** We consider several existing FL baselines such as FedAvg [40], $q$-FFL[31], CFFL [37], and an *extended contribution index* (ECI) method from [54] utilizing validation accuracy-based SV

Table 1: Average test accuracy (%) achieved by the agents collaborating via our fair gradient reward mechanism with varying degrees of altruism $\beta$ vs. tested baselines on all datasets. Each value in brackets denotes the highest test accuracy achieved by any agent.

| | MNIST | | | | | | CIFAR-10 | | | MR | SST |
|---|---|---|---|---|---|---|---|---|---|---|---|
| No. Agents | 10 | | | 20 | | | 10 | | | 5 | 5 |
| Data Partition | UNI | POW | CLA | UNI | POW | CLA | UNI | POW | CLA | POW | POW |
| Standalone | 91 (91) | 88 (92) | 53 (92) | 91 (91) | 89 (92) | 48 (90) | 46 (47) | 43 (49) | 31 (44) | 47(56) | 31(34) |
| FedAvg | 93 (94) | 92 (94) | 53 (93) | 93 (93) | 92 (94) | 49 (92) | 48 (48) | 47 (50) | 32 (47) | 51(63) | 33(35) |
| q-FFL | 85 (91) | 27 (45) | 44 (64) | 88 (91) | 48 (53) | 40 (59) | 41 (46) | 36 (36) | 22 (28) | 12(18) | 23(25) |
| CFFL | 90 (92) | 85 (90) | 34 (44) | 91 (93) | 88 (91) | 39 (46) | 39 (41) | 35 (45) | 22 (40) | 44(53) | 31(32) |
| ECI | 94 (94) | 92 (94) | 53 (94) | 94 (94) | 92 (94) | 49 (92) | 49 (49) | 47 (51) | 31 (46) | 56(61) | 33(34) |
| DW | 93 (94) | 92 (94) | 53 (93) | 93 (93) | 92 (94) | 49 (92) | 48 (48) | 47 (50) | 32 (47) | 51(62) | 33(35) |
| RR | 94 (95) | **95** (95) | 64 (72) | 94 (95) | 94 (95) | 50 (56) | 47 (59) | 49 (51) | 26 (29) | **63**(65) | **36**(36) |
| Ours (EU) | 94 (94) | 94 (94) | 54 (94) | 94 (94) | 94 (94) | 49 (92) | 49 (49) | 49 (51) | 32 (46) | 54(59) | 34(36) |
| Ours ($\beta = 1$) | 96 (97) | 94 (**95**) | 74 (**95**) | 95 (96) | 96 (**97**) | 65 (93) | 61 (**62**) | 60 (**62**) | 35 (**54**) | 62(**76**) | 35(36) |
| Ours ($\beta = 1.2$) | 94 (95) | **95** (95) | **75** (**95**) | 96 (96) | 96 (**97**) | 65 (93) | 61 (**62**) | 60 (**62**) | 35 (**54**) | 62(75) | 34(**37**) |
| Ours ($\beta = 1.5$) | **97** (**97**) | **95** (95) | **75** (**95**) | 96 (**97**) | 94 (95) | 65 (93) | 61 (**62**) | 59 (**62**) | 35 (**54**) | 62(74) | 35(**37**) |
| Ours ($\beta = 2$) | 96 (96) | **95** (**96**) | 73 (94) | **97** (**97**) | 95 (96) | **66** (**95**) | **62** (**62**) | **61** (**62**) | **36** (**54**) | 62(75) | 35(**37**) |

and setting $q_{i,t}$ for $i \in \mathcal{N}$ in (5) to be proportional to the agents' CIs. CFFL also utilizes the validation accuracy but is more efficient by using a leave-one-out approach instead of SV, while $q$-FFL aims at achieving egalitarian fairness by equalizing the local training losses of the agents. Furthermore, we implement simple FL baselines based on *round robin* (RR), *dataset weighted download* (DW), and *Euclidean distance* (EU). RR is commonly adopted in mechanism design to ensure fairness [6, 34] and also used in FL to schedule gradient downloads [51, 67]. For DW (EU), $q_{i,t}$ for $i \in \mathcal{N}$ in (5) are set to be proportional to the agents' local dataset sizes (negative Euclidean distance of their unnormalized parameter updates from that of the server). We also include *standalone* agents as a baseline, i.e., each agent trains its CNN using only its local dataset without involving FL.

**Performance Metrics.** To measure fairness, we consider the *scaled Pearson correlation coefficient*[3] $\rho := 100 \times \text{pearsonr}(\boldsymbol{\varphi}, \boldsymbol{\xi}) \in [-100, 100]$ between the test accuracies $\boldsymbol{\varphi}$ achieved by the agents when standalone [37] vs. that $\boldsymbol{\xi}$ achieved by them when collaborating via a gradient reward mechanism in FL after the entire training process has ended at iteration $t = T$. The corresponding experimental results will be reported in Sec. 4.2. To empirically verify the fairness guarantee in Theorem 2, we have also reported in Appendix B.3 results on the fairness metric $\rho$ between the importance coefficients $\boldsymbol{\varphi} := (r_{i,T})_{i=1,\ldots,N}$ (4) (i.e., measuring overall qualities of the parameter updates/gradients uploaded/contributed by the agents) vs. test accuracies (or negative training losses) $\boldsymbol{\xi}$ achieved by them. We consider other performance metrics like predictive performance (i.e., average and highest test accuracies achieved by the agents) and time overhead of the tested gradient reward mechanisms.

**Data Partitions among Agents.** We carefully construct two heterogeneous data partitions by varying the agents' local dataset sizes and corresponding numbers of distinct classes. For **imbalanced dataset sizes** (POW), we follow a power law to partition the entire dataset among the agents. For MNIST, we partition the entire dataset of size $\{3000, 6000, 12000\}$, respectively, among $\{5, 10, 20\}$ agents s.t. each agent has a randomly sampled local dataset of size 600 on average [40]. The size of the local dataset increases from the first to the last agent. Since the local dataset sizes vary significantly (i.e., superlinearly) among the agents, the agents with larger local datasets are expected to achieve better predictive performance. For **imbalanced class numbers** (CLA), we vary the number of distinct classes in the local datasets of the agents, while keeping their sizes fixed at 600. For this setting, we only consider MNIST and CIFAR-10 datasets and partition classes in a "linspace" manner as both contain 10 classes. To illustrate, for MNIST with 5 agents, agents $1, 2, 3, 4, 5$ own local datasets with $1, 3, 5, 7, 10$ classes, respectively; so, agent 1 (5) has a local dataset with 1 (10) class(es). Similarly, the agents with local datasets containing more classes are expected to achieve better predictive performance. We also include the simplest setting of the uniform/homogeneous data partition (UNI) where the agents are expected to achieve comparable predictive performance.

Additional details of the experimental settings are described in Appendix B.1.

## 4.2 Experimental Results

**Predictive Performance.** Table 1 shows results of the average and highest test accuracies achieved by the agents collaborating via our fair gradient reward mechanism vs. tested baselines on all

---

[3]The Pearson correlation coefficient has been applied to a similar use case in [19].

Table 2: Fairness metric $\rho \in [-100, 100]$ achieved by our fair gradient reward mechanism with varying degrees of altruism $\beta$ vs. tested baselines on all datasets. Higher value means greater fairness.

| | MNIST | | | | | | CIFAR-10 | | | MR | SST |
|---|---|---|---|---|---|---|---|---|---|---|---|
| No. Agents | 10 | | | 20 | | | 10 | | | 5 | 5 |
| Data Partition | UNI | POW | CLA | UNI | POW | CLA | UNI | POW | CLA | POW | POW |
| FedAvg | -45.60 | 55.24 | 24.12 | 0.85 | -32.58 | 40.83 | 18.47 | 97.48 | 98.75 | 48.68 | 57.50 |
| q-FFL | -44.73 | 39.00 | 22.38 | -22.01 | 38.71 | 48.07 | -17.64 | 51.33 | 94.06 | 56.43 | -75.92 |
| CFFL | 83.57 | 91.80 | 81.24 | 82.52 | 94.70 | 85.71 | 78.25 | 72.55 | 81.31 | 96.85 | 93.34 |
| ECI | 85.26 | **99.83** | **99.98** | 80.95 | **99.41** | 95.21 | 75.85 | 79.50 | 99.55 | 97.69 | 95.00 |
| DW | 89.15 | 98.93 | 65.34 | 86.94 | 99.63 | 35.21 | -23.14 | 91.97 | 45.45 | **99.20** | 97.12 |
| RR | 83.77 | 71.17 | -26.75 | -18.64 | 25.47 | 95.86 | 30.67 | 0.70 | 90.67 | 44.16 | -25.11 |
| Ours (EU) | 84.25 | 98.25 | 99.82 | 80.55 | 97.77 | **99.97** | 78.25 | 94.24 | 94.95 | 97.58 | 93.21 |
| Ours ($\beta = 1$) | 94.03 | 95.74 | 94.54 | 84.47 | 96.39 | 97.23 | **98.80** | **98.78** | **99.89** | 96.01 | 98.20 |
| Ours ($\beta = 1.2$) | 94.75 | 97.28 | 96.23 | 90.52 | 97.72 | 95.21 | 91.07 | 91.59 | 99.82 | 96.12 | **98.47** |
| Ours ($\beta = 1.5$) | **96.34** | 86.99 | 95.37 | 82.68 | 90.94 | 98.75 | 93.55 | 93.78 | 95.89 | 95.32 | 97.88 |
| Ours ($\beta = 2$) | 94.66 | 91.20 | 95.38 | **96.90** | 91.33 | 94.32 | 89.80 | 88.78 | 93.39 | 92.22 | 95.74 |

datasets. Our fair gradient reward mechanism generally outperforms the tested baselines on both metrics, especially for heterogeneous data partitions and on the MR dataset. On MNIST, for the CLA data partition among 10 agents, our fair gradient reward mechanism achieves average (highest) test accuracy of 75% (95%) at $\beta = 1.5$, while the best-performing ECI baseline achieves only that of 53% (94%). On CIFAR-10, for the CLA data partition among 10 agents, our fair gradient reward mechanism achieves average (highest) test accuracy of 36% (54%) at $\beta = 2$, while the best-performing DW baseline achieves only that of 32% (47%). On the MR dataset, our fair gradient reward mechanism achieves average (highest) test accuracy of 62% (76%) at $\beta = 1$, while the best-performing RR baseline achieves that of 63% (65%). Its better performance may be attributed to the adaptive re-weighting in the gradient aggregation step (1) via $r_{i,t}$, which can dynamically account for the heterogeneity in the agents' local datasets [32]. While EU performs comparably to both FedAvg and ECI (i.e., difference in average test accuracies between EU vs. FedAvg/ECI is less than 3%), it does not perform better than our fair gradient reward mechanism (e.g., on MNIST, for the CLA data partition among 10 agents, the difference in average test accuracies between EU vs. our fair gradient reward mechanism at $\beta = 1.5$ is more than 20%) because unlike cosine similarity, Euclidean distance fails to capture the directional difference between gradients, which is important since the negative gradients are pointing in the direction of lower loss. Importantly, $q$-FFL aims to equalize the local training losses w.r.t. the agent's local datasets, which may be suboptimal for heterogeneous data partitions like POW and CLA. We provide further results in Appendix B.5 empirically comparing the predictive performances of our fair gradient reward mechanism vs. $q$-FFL.

**Fairness.** To empirically verify the fairness guarantee in Theorem 2, Table 2 shows results on the fairness metric $\rho$ achieved by our fair gradient reward mechanism vs. tested baselines on all datasets. From Table 2, our fair gradient reward mechanism achieves a high degree of fairness of above 80, while the commonly used FedAvg performs suboptimally s.t. it produces the lowest degree of fairness of $-45.6$. On MNIST, for the POW data partition among 10/20 agents and the CLA data partition among 10 agents, ECI outperforms our fair gradient reward mechanism, albeit at a much higher time overhead of over 100 times and with additional information from an auxiliary dataset. CFFL underperforms our fair gradient reward mechanism and ECI as it adopts the leave-one-out approach which seems less accurate than SV in valuing the contributions of the agents [19]. Both $q$-FFL and RR promote egalitarian fairness instead of our notion of fairness via SV and hence do not perform optimally. DW achieves high degrees of fairness only for the POW data partition because it uses the agents' local dataset sizes to determine their gradient rewards. Fig. 3 illustrates an intuitive trend of the predictive performances achieved by 10 agents collaborating via our fair gradient reward mechanism for homogeneous and heterogeneous data partitions among the agents on MNIST and CIFAR-10: For the UNI data partition, all agents achieve comparable predictive performance. Their predictive performances vary more (most) for the POW (CLA) data partition, hence demonstrating that our fair gradient reward mechanism can distinguish the contributions of the agents and reward them with sparsified gradients fairly.

We have performed an additional experiment to understand our fair gradient reward mechanism for homogeneous and heterogeneous data partitions among 3 agents on MNIST and CIFAR-10 where for POW and CLA, agent 1 (3) uploads/contributes parameter updates/gradients of lowest (highest) quality over the entire training process. Fig. 4 shows how $r_{i,t}$ for agent $i = 1, 3$ varies over iterations $t$: Interestingly, for the CLA data partition, though agent 3 (brown solid line) is initially mistaken to

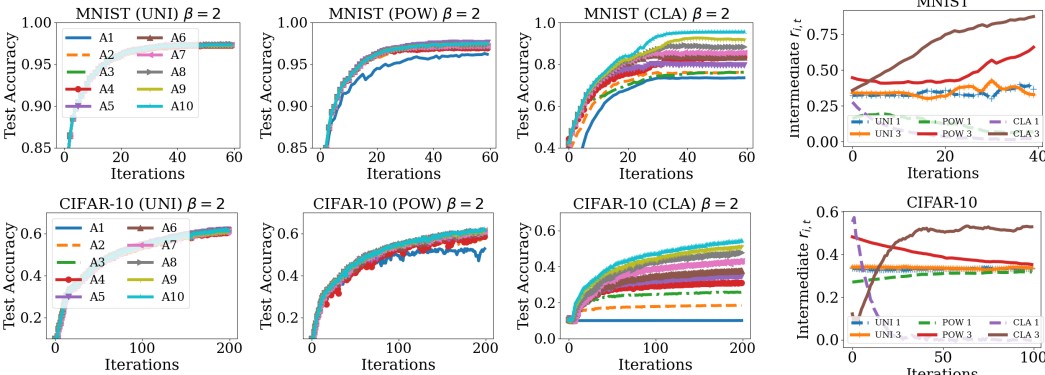

Figure 3: Test accuracy achieved by agent $i = 1, \ldots, 10$ (abbreviated to A$i$) collaborating via our fair gradient reward mechanism at $\beta = 2$ for the UNI (left), POW (middle), and CLA (right) data partitions among the 10 agents on MNIST (top) and CIFAR-10 (bottom). Their predictive performances vary least, more, and most for the respective UNI, POW, and CLA data partitions.

Figure 4: Graphs of $r_{i,t}$ (4) for agent $i = 1, 3$ vs. iteration $t$ for UNI, POW, and CLA data partitions among 3 agents on MNIST and CIFAR-10.

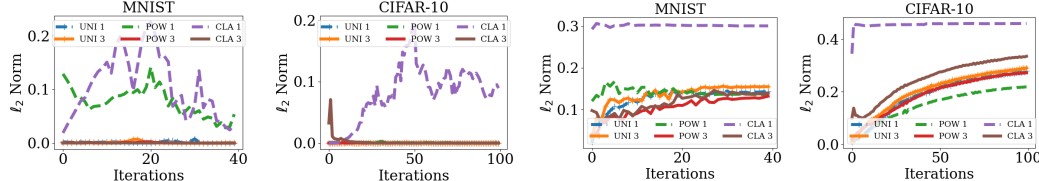

Figure 5: Graphs of $\ell_2$ distance between downloaded $\boldsymbol{v}_{i,t}$ (5) of agent $i = 1, 3$ and aggregated $\boldsymbol{u}_{\mathcal{N},t}$ (1) vs. iteration $t$ for UNI, POW, and CLA data partitions among 3 agents on MNIST (left) and CIFAR-10 (right).

Figure 6: Graphs of $\ell_2$ distance between last layer's model parameters of agent $i = 1, 3$ and that of the server vs. iteration $t$ for UNI, POW, and CLA data partitions among 3 agents on MNIST (left) and CIFAR-10 (right).

provide a low contribution, the dynamic update of $r_{3,t}$ (4) allows its true contribution to be recognized quickly. Fig. 5 (Fig. 6) shows how the $\ell_2$ distance between the downloaded sparsified gradient $\boldsymbol{v}_{i,t}$ (5) of agent $i = 1, 3$ and aggregated parameter update/gradient $\boldsymbol{u}_{\mathcal{N},t}$ (1) (last layer's model parameters of agent $i = 1, 3$ and that of the server) varies over iterations $t$: In particular, for the CLA data partition, agent $i = 1$ ($i = 3$) who uploads/contributes parameter updates/gradients of lowest (highest) quality over the entire training process downloads $\boldsymbol{v}_{i,t}$ as reward that is further from (closer to) $\boldsymbol{u}_{\mathcal{N},t}$, hence training last layer's model parameters to be further from (closer to) that of the server. Such results further validate that in Fig. 2 previously.

Lastly, Fig. 7 confirms that for the CLA data partition among 10 agents on MNIST, increasing the degree of altruism $\beta$ leads to all agents downloading higher-quality gradient rewards $\boldsymbol{v}_{i,t}$ (5) and thus incurring smaller training loss. In particular, agent 1 (abbreviated to A1 and represented by a blue solid line) who uploads/contributes parameter updates/gradients of lowest quality over the entire training process benefits most as $\beta$ increases, as explained previously in Sec. 3.4. Additional results w.r.t. test loss are reported in Appendix B.4.

**Time Overhead.** Table 3 compares the time overhead (seconds) of our fair gradient reward mechanism vs. tested baselines on all datasets; the ratio between the time overhead vs. training time is given in brackets. Our fair gradient reward mechanism is much more efficient than ECI and CFFL which also consistently achieve fairness. In particular, our fair gradient reward mechanism incurs a small time overhead of at most $0.4\times$ of the training time, while ECI incurs a significant time overhead of up to $140\times$ of the training time due to the calculation of the CI incurring $\mathcal{O}(2^N)$ time, even with the permutation sampling-based approximation [39, 56] for 10/20 agents. CFFL incurs at most $2\times$ of the training time (i.e., 5-6 times longer than ours) from the additional validation in each iteration.

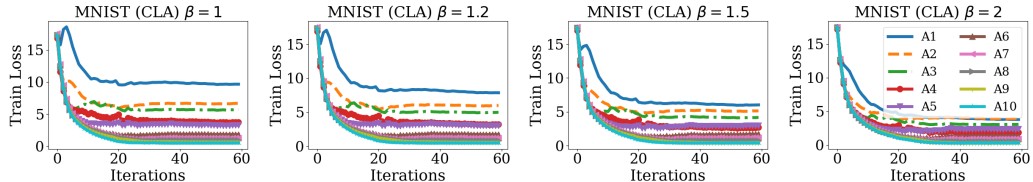

Figure 7: Training losses incurred by agent $i = 1, \ldots, 10$ (abbreviated to A$i$) collaborating via our fair gradient reward mechanism with varying degrees of altruism $\beta = 1.0, 1.2, 1.5, 2$ for the CLA data partition on MNIST.

Table 3: Time overhead (seconds) of our fair gradient reward mechanism vs. tested baselines on all datasets. Each value in brackets denotes the ratio between the time overhead vs. training time.

| | MNIST | | | CIFAR-10 | | MR | SST |
|---|---|---|---|---|---|---|---|
| No. Agents | 5 | 10 | 20 | 5 | 10 | 5 | 5 |
| FedAvg | 1.17 (7e-3) | 1.05 (1e-2) | 4.29 (1e-2) | 1.66 (7e-3) | 7.41 (1e-2) | 1.3 (1e-4) | 1.31 (6e-4) |
| q-FFL | 6.14 (4e-2) | 4.97 (5e-2) | 91.20 (0.3) | 97.28 (0.4) | 58.94 (7e-2) | 90.01 (8e-3) | 82.85 (4e-2) |
| CFFL | 32.15 (0.2) | 21.79 (0.3) | 500.03 (1.6) | 570.12 (2.0) | 302.44 (0.4) | 479.12 (0.2) | 487.71 (2e-1) |
| ECI | 2377.33 (16) | 11937.80 (141) | 23749.06 (74) | 3571.75 (15) | 58835.83 (84) | 422.85 (4e-2) | 801.20 (0.4) |
| DW | **0.89 (6e-3)** | **0.79 (9e-3)** | **1.60 (5e-3)** | **1.21 (5e-3)** | **5.29 (7e-3)** | **0.99 (1e-5)** | 0.98 (5e-4) |
| RR | **0.89 (6e-3)** | 0.82 (9e-3) | **1.60 (5e-3)** | 3.31 (1e-2) | 5.41 (7e-3) | 1.01 (5e-4) | **0.99 (5e-4)** |
| Ours (EU) | **0.89 (6e-3)** | 0.81 (9e-3) | 1.61 (5e-3) | 1.22 (5e-3) | 5.33 (7e-3) | 1.01 (5e-4) | **0.99 (5e-4)** |
| Ours (Cosine) | 6.34 (4e-2) | 4.94 (5e-2) | 94.30 (0.3) | 98.39 (0.4) | 54.94 (7e-2) | 89.81 (8e-3) | 82.87 (4e-2) |

**Hyperparameters.** We find that $\alpha \in [0.8, 1)$(i.e., relative weight on $r_{i,t-1}$ in (4)), $\beta \in [1, 2]$ (i.e., degree of altruism in (5)) and $\Gamma \in [0.1, 1]$ (i.e., normalization coefficient in (1)) are effective in achieving competitive predictive performance and fairness. In our experiments, we set $\alpha = 0.95$, $\beta = [1, 1.2, 1.5, 2]$, and $\Gamma = 0.5$ for MNIST, $\Gamma = 0.15$ for CIFAR-10, and $\Gamma = 1$ for SST and MR.

## 5    Conclusion and Future Work

In this paper, we have described a novel *cosine gradient Shapley value* (CGSV) (Sec. 3.2) to fairly evaluate the expected marginal contribution of each agent's uploaded model parameter update/gradient in FL without needing an auxiliary validation dataset and present an efficient approximation of CGSV with a bounded error (Sec. 3.3). Based on the approximate CGSV, we have designed a novel training-time fair gradient reward mechanism (Sec. 3.4) by exploiting the trick of sparsifying the aggregated parameter update/gradient downloaded from the server as reward to each agent such that its resulting quality is commensurate to that of the agent's uploaded/contributed parameter update/gradient. Consequently, an agent who uploads/contributes higher-quality parameter updates/gradients over the entire training process should eventually be rewarded with converged model parameters whose resulting training loss (and hence predictive performance) is closer to that of the server, as demonstrated in our fairness guarantee (Sec. 3.5). We have empirically demonstrated the effectiveness of our fair gradient reward mechanism on multiple benchmark datasets in terms of fairness, predictive performance, and time overhead (Sec. 4). In particular, our fair gradient reward mechanism is much more efficient than several existing FL baselines since it requires only slight calculations by the server.

Our proposed fair gradient reward mechanism also provides practitioners the flexibility to trade off between fairness and equality in gradient rewards via a hyperparameter $\beta$ controlling the degree of altruism (Sec. 3.4). For future work, it would be interesting to consider the notion of fairness when there are some adversaries. We would also consider generalizing our work and fairness guarantee to other types of CML (e.g., model fusion [16, 17, 24]) and collaborative Bayesian optimization [53].

## Acknowledgments and Disclosure of Funding

This research is supported by the National Research Foundation, Singapore under its AI Singapore Programme (Award No: AISG2-RP-2020-018). Any opinions, findings and conclusions or recommendations expressed in this material are those of the author(s) and do not reflect the views of National Research Foundation, Singapore. Xinyi Xu is supported by the Institute for Infocomm Research of Agency for Science, Technology and Research (A*STAR).

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
