## A Theoretical Results

### A.1 Fairness Properties of CGSV

For $\mathcal{S} \subseteq \mathcal{N} \setminus \{i\}$, let $\Delta_{\mathcal{S},i} := \nu(\mathcal{S} \cup i) - \nu(\mathcal{S})$, the following properties are satisfied by CGSV:

(P1) Null Player [58]: $\Delta_{\mathcal{S},i} = 0, \forall \mathcal{S} \subseteq \mathcal{N} \setminus \{i\} \implies \phi_i = 0$.

(P2) Symmetry [58]: $\Delta_{\mathcal{S},i} = \Delta_{\mathcal{S},i'}, \forall \mathcal{S} \subseteq \mathcal{N} \setminus \{i,i'\} \implies \phi_i = \phi_{i'}$.

(P3) Strict Desirability [3]: $\Delta_{\mathcal{S},i} \geq \Delta_{\mathcal{S},i'}, \forall \mathcal{S} \subseteq \mathcal{N} \setminus \{i,i'\}$ and $\exists \mathcal{S}' \subseteq \mathcal{N} \setminus \{i,i'\}$ s.t. $\Delta_{\mathcal{S}',i} > \Delta_{\mathcal{S}',i'} \implies \phi_i > \phi_{i'}$.

(P4) Coalitional Monotonicity [64, Equ.(4)]: $\nu(\mathcal{S}) \geq \nu'(\mathcal{S})$ for some $\mathcal{S} \subseteq \mathcal{N}$ and $\nu'(\mathcal{S}') = \nu'(\mathcal{S}') \forall \mathcal{S}' \subseteq \mathcal{N} \ \mathcal{S}' \neq \mathcal{S} \implies \phi_i(\nu) \geq \phi_i(\nu') \ \forall i \in \mathcal{S}$.

(P5) Individual Monotonicity [64, Equ.(5)]: $\forall i, \nu(), \nu'()$, $\nu(\mathcal{S}) \geq \nu'(\mathcal{S}) \ \forall \mathcal{S}$ containing $i$ and $\nu(\mathcal{S}') = \nu'(\mathcal{S}') \ \forall \mathcal{S}'$ not containing $i \implies \phi_i(\nu) \geq \phi_i(\nu')$.

(P1) can be intuitively understood as if $i$ adds zero value to the group, then the corresponding CGSV will be zero. This is to prevent agents who wish to exploit the system by uploading randomly generated gradients instead of actual gradients. Note that in a high-dimensional space, the cosine similarity between a random gradient and an actual gradient is likely to be close to zero.

(P2) and (P3) provide a comparative relationship between any pair of agents $i, i'$. In the simplest case as in (P2), $i, i'$ provide exactly identical contributions, then their corresponding CGSVs are equal. On the other hand, if $i$ consistently provides more than $i'$ as in (P3), then the CGSV for $i$ is higher to correctly reflect this relation. Therefore, these two properties ensure the agents who contribute more by uploading better gradients are properly recognized (with higher CGSV). This is crucial in our designed reward mechanism which follows such relations in the CGSV.

(P4) states that if a group of agents in $\mathcal{S}$ collectively do better, while all other groups $\mathcal{S}' \neq \mathcal{S}$ stay the same, then the agents in $\mathcal{S}$ do not lose. In particular, applying (P4) repeatedly gives an equivalent result regarding a single agent $i$, which we call *individual monotonicity* as in (P5).

(P5) takes the perspective of agent $i$ while all other agents do not change, and agent $i$ makes better contributions and improves (or at least does not hurt) all the coalitions $i$ is in, then agent $i$ does not lose. Consequently, it implies an incentive for the agents to make better contributions which could increase their CGSVs, which in turn correspond to better rewards in our mechanism.

### A.2 Proof of Theorem 1

Theorem 1 relies on the following equivalent formulation of $\phi_i$,

$$\phi_i = \underbrace{\sum_{\mathcal{S} \subseteq \mathcal{N} \setminus \{i\}} A_{\mathcal{S}} \nu(\mathcal{S})}_{\text{additive error}} + [\underbrace{\sum_{\mathcal{S} \subseteq \mathcal{N} \setminus \{i\}} B_{\mathcal{S}}}_{\text{multiplicative factor } L_i}] \ \psi_i \qquad (6)$$

where $A_{\mathcal{S}}, B_{\mathcal{S}}$ are constants specific to $\mathcal{S}$. Note that Theorem 1 provides an upper bound for the additive error in (6) so we approximate $\phi_i \approx L_i \psi_i$. Further, by recalling a property of CGSV (invari-

ance under linear transformation), we can avoid explicitly calculating the multiplicative factor $L_i$ via normalization of $\psi_i$ if all $L_i$ are approximately equal, as we show in Lemma 1.

$$\phi_i = \underbrace{\sum_{\mathcal{S} \subseteq \mathcal{N} \setminus \{i\}} A_{\mathcal{S}} \nu(\mathcal{S})}_{\text{additive error}} + \underbrace{\left[ \sum_{\mathcal{S} \subseteq \mathcal{N} \setminus \{i\}} B_{\mathcal{S}} \right]}_{\text{multiplicative factor } L_i} \psi_i$$

Before the proof, we first show the derivation (reproduced above). The intuitive idea is that the approximation $\psi_i := \cos(\boldsymbol{u}_i, \boldsymbol{u}_{\mathcal{N}})$ appears in the summation of $\phi_i$ repeatedly, so we collect all its coefficients into the multiplicative factor $L_i$ and collect everything else as the additive error.

$$\phi_i = \frac{1}{N} \sum_{\mathcal{S} \subseteq \mathcal{N} \setminus \{i\}} \frac{1}{\binom{N-1}{|\mathcal{S}|}} \nu(\mathcal{S} \cup \{i\}) - \nu(\mathcal{S})$$

$$= \frac{1}{N} \sum_{\mathcal{S} \subseteq \mathcal{N} \setminus \{i\}} \frac{1}{\binom{N-1}{|\mathcal{S}|}} \underbrace{(\cos(\boldsymbol{u}_{\mathcal{S} \cup \{i\}}, \boldsymbol{u}_{\mathcal{N}}) - \cos(\boldsymbol{u}_{\mathcal{S}}, \boldsymbol{u}_{\mathcal{N}}))}_{A}$$

We focus on $A$ and leave the rest unchanged.

$$A = \frac{\langle \boldsymbol{u}_{\mathcal{S} \cup \{i\}}, \boldsymbol{u}_{\mathcal{N}} \rangle}{\|\boldsymbol{u}_{\mathcal{S} \cup \{i\}}\| \times \|\boldsymbol{u}_{\mathcal{N}}\|} - \frac{\langle \boldsymbol{u}_{\mathcal{S}}, \boldsymbol{u}_{\mathcal{N}} \rangle}{\|\boldsymbol{u}_{\mathcal{S}}\| \times \|\boldsymbol{u}_{\mathcal{N}}\|}$$

$$= \frac{\langle \boldsymbol{u}_{\mathcal{S} \cup \{i\}}, \boldsymbol{u}_{\mathcal{N}} \rangle}{\Gamma_1 \Gamma_{\mathcal{N}}} - \frac{\langle \boldsymbol{u}_{\mathcal{S}}, \boldsymbol{u}_{\mathcal{N}} \rangle}{\Gamma_2 \Gamma_{\mathcal{N}}}$$

$$= \frac{1}{\Gamma_1 \Gamma_2 \Gamma_{\mathcal{N}}} \langle \Gamma_2 \boldsymbol{u}_{\mathcal{S} \cup \{i\}} - \Gamma_1 \boldsymbol{u}_{\mathcal{S}}, \boldsymbol{u}_{\mathcal{N}} \rangle$$

$$= \frac{(\Gamma_2 - \Gamma_1) \langle \boldsymbol{u}_{\mathcal{S}}, \boldsymbol{u}_{\mathcal{N}} \rangle}{\Gamma_1 \Gamma_2 \Gamma_{\mathcal{N}}} - \frac{r_i \Gamma_2 \langle \boldsymbol{u}_i, \boldsymbol{u}_{\mathcal{N}} \rangle}{\Gamma_1 \Gamma_2 \Gamma_{\mathcal{N}}}$$

$$= \frac{\Gamma_2 - \Gamma_1}{\Gamma_1} \nu(\mathcal{S}) - \frac{r_i \Gamma}{\Gamma_1} \cos(\boldsymbol{u}_i, \boldsymbol{u}_{\mathcal{N}})$$

where $\Gamma_1 := \|\boldsymbol{u}_{\mathcal{S} \cup \{i\}}\|$, $\Gamma_2 := \|\boldsymbol{u}_{\mathcal{S}}\|$ and $\Gamma_{\mathcal{N}} = \|\boldsymbol{u}_{\mathcal{N}}\|$. Substitute this back with $A_{\mathcal{S}} = \frac{1}{N} \frac{1}{\binom{N-1}{|\mathcal{S}|}} \frac{\Gamma_2 - \Gamma_1}{\Gamma_1}$ and $B_{\mathcal{S}} = \frac{1}{N} \frac{1}{\binom{N-1}{|\mathcal{S}|}} \frac{r_i \Gamma}{\Gamma_1}$ to complete the derivation.

*Proof Sketch of Theorem 1.* The high-level idea is that, with cosine similarity, the approximation $\psi_i := \cos(\boldsymbol{u}_i, \boldsymbol{u}_{\mathcal{N}})$ is a significant component of the actual CGSV $\phi_i$ by a multiplicative factor. Because the multiplicative coefficient $\frac{1}{\binom{N-1}{|\mathcal{S}|}}$ becomes very small with a large $N$, so it reduces the effect of the terms *not* involving $\boldsymbol{u}_i$. While it also reduces the effect of the terms involving $\boldsymbol{u}_i$, the idea is that if the actual contribution from $\boldsymbol{u}_i$ is relatively large (by the assumption $|\langle r_i \boldsymbol{u}_i, \boldsymbol{u}_{\mathcal{N}} \rangle| \geq \frac{1}{I}$), then we ensure the error is small relatively. Note in the theorem, we have absorbed $r_i$ into the constant $\frac{1}{I}$. $\qquad\square$

*Proof of Theorem 1.* Notice the summation enumerates the same list of terms for both, so we minimize $A_{\mathcal{S}} \nu(\mathcal{S})$ relative to $B_{\mathcal{S}} \psi_i$. Specifically, we examine the pairwise ratio between the two corresponding terms $\frac{\Gamma_2 - \Gamma_1}{\Gamma_1} \cos(\boldsymbol{u}_{\mathcal{S}}, \boldsymbol{u}_{\mathcal{N}})$ and $\frac{r_i \Gamma}{\Gamma_1} \cos(\boldsymbol{u}_i, \boldsymbol{u}_{\mathcal{N}})$ in the summation as follows:

$$\frac{(|\Gamma_2 - \Gamma_1) \cos(\boldsymbol{u}_{\mathcal{S}}, \boldsymbol{u}_{\mathcal{N}})|}{|r_i \Gamma \cos(\boldsymbol{u}_i, \boldsymbol{u}_{\mathcal{N}})|} = \frac{|\Gamma_2 - \Gamma_1|}{|\Gamma_2|} \frac{|\langle \boldsymbol{u}_{\mathcal{S}}, \boldsymbol{u}_{\mathcal{N}} \rangle|}{|r_i \langle \boldsymbol{u}_i, \boldsymbol{u}_{\mathcal{N}} \rangle|}$$

$$= |\Gamma_2 - \Gamma_1| \frac{|\langle \boldsymbol{u}_{\mathcal{S}}, \boldsymbol{u}_{\mathcal{N}} \rangle|}{|\Gamma_2|} \frac{1}{|r_i \langle \boldsymbol{u}_i, \boldsymbol{u}_{\mathcal{N}} \rangle|}$$

$$\leq \Gamma \sqrt{\|\frac{\boldsymbol{u}_{\mathcal{S}}}{\Gamma_2}\|^2 \|\boldsymbol{u}_{\mathcal{N}}\|^2} \frac{1}{r_i |\langle \boldsymbol{u}_i, \boldsymbol{u}_{\mathcal{N}} \rangle|}$$

$$\leq \Gamma \Gamma_{\mathcal{N}} \frac{1}{r_i |\langle \boldsymbol{u}_i, \boldsymbol{u}_{\mathcal{N}} \rangle|}$$

$$\leq I \Gamma^2$$

We bound $|\Gamma_2 - \Gamma_1| \leq \Gamma$ with the gradient normalization constant $\Gamma$ by triangle inequality.

We bound $\langle \boldsymbol{u}_\mathcal{S}, \boldsymbol{u}_\mathcal{N} \rangle / \Gamma_2$ using Cauchy-Schwarz inequality, bound $\Gamma_\mathcal{N}$ with $\Gamma$ as $\boldsymbol{u}_\mathcal{N}$ is a convex sum of vectors each with norm $\Gamma$, and use the assumption to bound $1/r_i |\langle \boldsymbol{u}_i, \boldsymbol{u}_\mathcal{N} \rangle|$.

The above inequality bounds error-to-approximation ratio, i.e., $|A_\mathcal{S} \nu(S)|/|B_\mathcal{S} \psi_i|$ is bounded by $I\Gamma^2$ for every coalition $\mathcal{S}$ in the summation, which implies

$$\frac{|\sum_{\mathcal{S} \subseteq \mathcal{N} \setminus \{i\}} A_\mathcal{S} \nu(S)|}{|\sum_{\mathcal{S} \subseteq \mathcal{N} \setminus \{i\}} B_\mathcal{S} \psi_i|} \leq I\Gamma^2.$$

This result is useful because the error-to-approximation ratio is consistently bounded regardless of the normalization on $L_i \psi_i$, such as linear scaling we conduct subsequently.

If we additionally assume,

$$\frac{r_i \Gamma}{\Gamma_1} = \frac{r_i \|\boldsymbol{u}_i\|}{\|\sum_{i' \in \mathcal{S} \cup \{i\}} r_{i'} \boldsymbol{u}_{i'}\|} \leq 1,$$

then the error term $\phi_i - L_i \psi_i \leq I\Gamma^2$ before normalization. Its proof is by showing $|\sum_{\mathcal{S} \subseteq \mathcal{N} \setminus \{i\}} B_\mathcal{S} \psi_i \leq 1|$ and rearranging the terms in the previous inequality.

Note $|\psi_i| \leq 1$ and by the assumption $r_i \Gamma / \Gamma_1 \leq 1$, we first have $|r_i \Gamma \psi_i / \Gamma_1| \leq 1$, so $\sum_{\mathcal{S} \subseteq \mathcal{N} \setminus \{i\}} r_i \Gamma \psi_i / \Gamma_1 \leq \sum_{\mathcal{S} \subseteq \mathcal{N} \setminus \{i\}} 1 =$ number of terms in the summation. Putting back in the coefficients we can show that $\sum_{\mathcal{S} \subseteq \mathcal{N} \setminus \{i\}} 1/N \times 1/\binom{N-1}{|\mathcal{S}|} < 1$ because the enumeration $\mathcal{S} \subseteq \mathcal{N} \setminus \{i\}$ is not exhaustive while the coefficients are specified to have sum is 1 when the enumeration is exhaustive.

This additional assumption excludes degenerate cases where multiple agents (with approximately equal $r_i$'s) upload gradients in opposite directions and counteract each other which results in a net gradient vector approximately equal to a zero vector. Such cases are unlikely as the gradient vectors are calculated based on on randomly selected mini-batches, and these gradient vectors are in a high dimension.

$\square$

In order to recall the property that CGSV is invariant under linear transformation, we require that all $L_i$'s are approximately equal. To show this, we specify an assumption to exclude the degenerate cases by requiring $\boldsymbol{u}_{\mathcal{S} \cup \{i\}}$ and $\boldsymbol{u}_{\mathcal{S} \cup \{j\}}$ are lower bounded linearly in $\Gamma$. This assumption stipulates that $\boldsymbol{u}_{\mathcal{S} \cup \{i\}}$ and $\boldsymbol{u}_{\mathcal{S} \cup \{j\}}$ are away from zero vectors, and have norms of the same magnitude of $\boldsymbol{u}_i$ and $\boldsymbol{u}_j$.

**Lemma 1 (Closeness of $L_i$).** *Assume $\exists M > 0, s.t. \forall \mathcal{S} \subseteq \mathcal{N} \setminus \{i, i'\}, \min(\|\boldsymbol{u}_{\mathcal{S} \cup \{i\}}\|, \|\boldsymbol{u}_{\mathcal{S} \cup \{i'\}}\|) \geq M\Gamma$, then*

$$\max_{i, i' \in \mathcal{N}} L_i - L_{i'} \leq \sum_{s \in \mathcal{N} \setminus \{i, i'\}} \frac{1}{\binom{N-1}{|\mathcal{S}|}} \frac{2}{M^2 \Gamma}.$$

*Proof of Lemma 1.* Due to symmetry, $\boldsymbol{u}_i = \boldsymbol{u}_{i'} \implies L_i = L_{i'}$. We only need to consider $\boldsymbol{u}_i \neq \boldsymbol{u}_{i'}$.

We consider the terms by grouping the coalitions $\mathcal{S}$ into three types: 1) $i \notin \mathcal{S} \wedge i' \notin \mathcal{S}$; 2) $i \in \mathcal{S} \oplus i' \in \mathcal{S}$; 3) $i \in \mathcal{S} \wedge i' \in \mathcal{S}$. We need not consider 3) as the summation for $i$ is over $\mathcal{S} \subseteq \mathcal{N} \setminus \{i\}$.

For 2), let $\mathcal{S} \subseteq \mathcal{N} \setminus \{i, i'\}$ and $\mathcal{S}_i = \mathcal{S} \cup \{i\}, \mathcal{S}_j = \mathcal{S} \cup \{i'\}$. We can see that $\mathcal{S}_i, \mathcal{S}_{i'}$ constitute a pair of symmetric case for two terms in the summation of $L_i$ and $L_{i'}$ respectively. In particular, for $L_i$, the term in summation is $1/\binom{N-1}{|\mathcal{S}_{i'}|} \times 1/\|\boldsymbol{u}_{\mathcal{S}_{i'} \cup \{i\}}\|$. Since $\boldsymbol{u}_{\mathcal{S}_i \cup \{i'\}} = \boldsymbol{u}_{\mathcal{S} \cup \{i, i'\}} = \boldsymbol{u}_{\mathcal{S}_{i'} \cup \{i\}}$ and $|\mathcal{S}_i| = |\mathcal{S}| + 1 = |\mathcal{S}_{i'}|$, these two symmetric terms cancel out and the 2) type coalitions contribute exactly 0 to $L_i - L_{i'}$.

Now for 1) the coalitions $\mathcal{S} \subseteq \mathcal{N} \setminus \{i, i'\}$, we bound the sum of terms as follows,

$$L_i - L_j = \sum_{\mathcal{S} \subseteq \mathcal{N} \setminus \{i, i'\}} \frac{1}{\binom{N-1}{|\mathcal{S}|}} \left[ \frac{1}{\|\boldsymbol{u}_{\mathcal{S} \cup \{i\}}\|} - \frac{1}{\|\boldsymbol{u}_{\mathcal{S} \cup \{i'\}}\|} \right]$$

$$= \sum_{\mathcal{S} \subseteq \mathcal{N} \setminus \{i, i'\}} \frac{1}{\binom{N-1}{|\mathcal{S}|}} \frac{\|\boldsymbol{u}_{\mathcal{S} \cup \{i'\}}\| - \|\boldsymbol{u}_{\mathcal{S} \cup \{i\}}\|}{\|\boldsymbol{u}_{\mathcal{S} \cup \{i\}}\| \times \|\boldsymbol{u}_{\mathcal{S} \cup \{i'\}}\|}$$

$$\leq \sum_{\mathcal{S} \subseteq \mathcal{N} \setminus \{i, i'\}} \frac{1}{\binom{N-1}{|\mathcal{S}|}} \frac{2\Gamma}{M^2 \Gamma^2}$$

$$\leq \sum_{\mathcal{S} \subseteq \mathcal{N} \setminus \{i, i'\}} \frac{1}{\binom{N-1}{|\mathcal{S}|}} \frac{2}{M^2 \Gamma}.$$

The first inequality is because the numerator is upper bounded by $2\Gamma$ due to triangle inequality, and the denominator is lower bounded by $M^2 \Gamma^2$ using the assumption. This error decreases quickly with more agents due to the coefficient $\binom{N-1}{|\mathcal{S}|}^{-1}$. $\qquad \square$

### A.3   Proof of Theorem 2

This proof includes an intermediate step of showing $\delta_{i',t} \geq \delta_{i,t}$. First observe the following inequalities using the triangle inequality:

$$\delta_{i,t} \leq \delta_{i,t-1} + \|\boldsymbol{v}_{i,t}\| \quad \text{and} \quad \delta_{i',t} \geq \delta_{i',t-1} - \|\boldsymbol{v}_{i',t}\|. \tag{7}$$

From the condition

$$\delta_{i',t-1} - \delta_{i,t-1} \geq 2\|\boldsymbol{v}_{i,t}\|,$$

we have

$$\delta_{i',t-1} - \delta_{i,t-1} \geq 2\|\boldsymbol{v}_{i,t}\| \geq \|\boldsymbol{v}_{i,t}\| + \|\boldsymbol{v}_{i',t}\| \tag{8}$$

The inequality $\|\boldsymbol{v}_{i,t}\| \geq \|\boldsymbol{v}_{i',t}\|$ follows directly by applying $r_{i,t} \geq r_{i',t}$ to (5) and observing $\text{mask}(\cdot)$ retains the largest components in magnitude and making the rest zeros.

Rearranging (8) gives

$$\delta_{i',t-1} - \|\boldsymbol{v}_{i',t}\| \geq \delta_{i,t-1} + \|\boldsymbol{v}_{i,t}\|.$$

Connecting both inequalities in (7) gives

$$\delta_{i',t} \geq \delta_{i',t-1} - \|\boldsymbol{v}_{i',t}\| \geq \delta_{i,t-1} + \|\boldsymbol{v}_{i,t}\| \geq \delta_{i,t}.$$

Subsequently, we use $\delta_{i',t} \geq \delta_{i,t}$ and some regularity conditions of $\mathbf{F}()$ to establish $\mathbf{F}(\boldsymbol{w}_{i,t}) \leq \mathbf{F}(\boldsymbol{w}_{i',t})$. Specifically, we assume $\mathbf{F}()$ is both $L$-smooth and $\mu$-strongly convex with $L \leq \mu$.

We first recall the respective definitions for $\mu$-strongly convex and $L$-smooth functions.

**Definition 2** ($L$-**Smooth F**). If $\mathbf{F}$ is $L$-smooth, then $\forall \boldsymbol{w}, \boldsymbol{w}' \in \mathcal{W}$,

$$\mathbf{F}(\boldsymbol{w}) \leq \mathbf{F}(\boldsymbol{w}') + \nabla \mathbf{F}(\boldsymbol{w}')^T (\boldsymbol{w} - \boldsymbol{w}') + \frac{L}{2} \|\boldsymbol{w} - \boldsymbol{w}'\|^2.$$

**Definition 3** ($\mu$-**Strongly Convex F**). If $\mathbf{F}$ is $\mu$-strongly convex, then $\forall \boldsymbol{w}, \boldsymbol{w}' \in \mathcal{W}$,

$$\mathbf{F}(\boldsymbol{w}) \geq \mathbf{F}(\boldsymbol{w}') + \nabla \mathbf{F}(\boldsymbol{w}')^T (\boldsymbol{w} - \boldsymbol{w}') + \frac{\mu}{2} \|\boldsymbol{w} - \boldsymbol{w}'\|^2.$$

From $L$-smoothness, we have

$$\mathbf{F}(\boldsymbol{w}_{i,t}) \leq \underbrace{\mathbf{F}(\boldsymbol{w}_{\mathcal{N},t}) + \nabla \mathbf{F}(\boldsymbol{w}_{\mathcal{N},t})^\top (\boldsymbol{w}_{i,t} - \boldsymbol{w}_{\mathcal{N},t}) + \frac{L}{2} \delta_{i,t}^2}_{R_L}.$$

From $\mu$-strong convexity, we have

$$\mathbf{F}(\boldsymbol{w}_{i',t}) \geq \underbrace{\mathbf{F}(\boldsymbol{w}_{\mathcal{N},t}) + \nabla \mathbf{F}(\boldsymbol{w}_{\mathcal{N},t})^\top (\boldsymbol{w}_{i',t} - \boldsymbol{w}_{\mathcal{N},t}) + \frac{\mu}{2} \delta_{i',t}^2}_{R_\mu}.$$

In order to prove $\mathbf{F}(\boldsymbol{w}_{i,t}) \leq \mathbf{F}(\boldsymbol{w}_{i',t})$, it suffices to prove $R_L \leq R_\mu$ or equivalently $R_L - R_\mu \leq 0$.

$$R_L - R_\mu = \underbrace{\nabla\mathbf{F}(\boldsymbol{w}_{\mathcal{N},t})^\top(\boldsymbol{w}_{i,t} - \boldsymbol{w}_{i',t})}_{R_1} + \underbrace{\frac{1}{2}(L\delta_{i,t}^2 - \mu\delta_{i',t}^2)}_{R_2}.$$

With $L \leq \mu$ and $\delta_{i,t} \leq \delta_{i',t}$, we have

$$R_2 = \frac{1}{2}(L\delta_{i,t}^2 - \mu\delta_{i',t}^2) \leq \frac{L}{2}(\delta_{i,t}^2 - \delta_{i',t}^2) \leq 0.$$

We formalize $\boldsymbol{w}_{\mathcal{N},t}$ being near to a stationary point by specifying an upper bound on the gradient:

$$\|\nabla\mathbf{F}(\boldsymbol{w}_{\mathcal{N},t})\| \leq \frac{L|\delta_{i,t}^2 - \delta_{i',t}^2|}{2\|\boldsymbol{w}_{i,t} - \boldsymbol{w}_{i',t}\|}.$$

We have the following:

$$|R_1| \triangleq |\nabla\mathbf{F}(\boldsymbol{w}_{\mathcal{N},t})^\top(\boldsymbol{w}_{i,t} - \boldsymbol{w}_{i',t})| \leq \|\nabla\mathbf{F}(\boldsymbol{w}_{\mathcal{N},t})\| \times \|(\boldsymbol{w}_{i,t} - \boldsymbol{w}_{i',t})\|$$
$$\leq \frac{L|\delta_{i,t}^2 - \delta_{i',t}^2|}{2}$$
$$\leq |R_2|$$

where the first inequality is by Cauchy-Schwarz, the second inequality is by substituting the above upper bound and the last inequality is due to taking absolute values of two negative values.

Finally, since $|R_1| \leq |R_2|$ and $R_2 \leq 0$, we get $R_1 + R_2 \leq 0$ and hence $R_L + R_\mu \triangleq R_1 + R_2 \leq 0$.

## B  Experimental Results

### B.1  Experimental Settings

**Additional Details.** For CIFAR-10, we follow power law to randomly partition total $\{10000, 20000\}$ examples among $\{5, 10\}$ agents respectively. For MR (SST), we follow power law to randomly partition 9596 (8544) examples among 5 agents. We provide the training hyper-parameters used for different datasets in Table 4.

Table 4: Framework-independent hyper-parameters. Batch size $B$, initial step-size $\eta$, step-size exponential decay $\gamma$, total iterations $T$. Note for experiments with more than 5 agents for MNIST and CIFAR-10, $\eta$ is 0.25 and 0.025, respectively.

| Dataset | $B$ | $\eta\ (\gamma)$ | $T$ |
|---------|-----|------------------|-----|
| MNIST | 32 | 0.15 (0.977) | 60 |
| CIFAR-10 | 128 | 0.015 (0.977) | 200 |
| MR | 128 | 5e-5 (0.977) | 100 |
| SST | 256 | 1e-5 (0.977) | 100 |

**Experiment Hardware and Software.** All experiments are conducted on a server with 16 cores (Intel(R) Xeon(R) CPU E5-2683 v4 @ 2.10GHz), 256 GB RAM and 4 GPUs (GeForce GTX 1080 Ti). Our implementation mainly uses PyTorch, torchtext, torchvision and some auxiliary packages such as Numpy, Pandas and Matplotlib. The specific versions and package requirements are provided together with the source code. To reduce the impact of randomness in the experiments, we adopt several measures: fix the model initilizations (we initialize model weights and save them for future experiments); fix all the random seeds; and invoke the deterministic behavior of PyTorch. As a result, given the same model initialization, our implementation is expected to produce consistent results on the same machine over experimental runs.

### B.2  5-Agent Case for MNIST and CIFAR-10

For completion, we include the accuracy and fairness results under the consistent setting as the main paper for the 5-agent case for MNIST and CIFAR-10 for the three data partitions in Table 5 and Table 6, respectively.

Table 5: Average test accuracies (%) of all the agents for all baselines and our method with varying degrees of altruism $\beta$, on all four datasets. Values in the bracket denote the highest test accuracies among all the agents.

| Data Partition | MNIST $N = 5$ | | | CIFAR-10 $N = 5$ | | |
|---|---|---|---|---|---|---|
| | UNI | POW | CLA | UNI | POW | CLA |
| *Standalone* | 91(91) | 87(94) | 50(91) | 44(46) | 42(52) | 29(44) |
| *FedAvg* | 93(93) | 91(95) | 50(92) | 46(47) | 46(52) | 30(45) |
| q-FFL | 82(85) | 59(78) | 49(84) | 31(32) | 31(34) | 19(24) |
| CFFL | 24(39) | 21(37) | 27(28) | 44(45) | 40(49) | 26(43) |
| *ECI* | 93(94) | 94(95) | 52(92) | 46(47) | 44(44) | 30(41) |
| *DW* | 93(93) | 91(95) | 50(92) | 46(47) | 46(52) | 30(45) |
| *RR* | 94(95) | 95(95) | 67(73) | 40(45) | 49(57) | 23(32) |
| *Ours (EU)* | 94(94) | 93(95) | 50(92) | 47(48) | 48(52) | 30(45) |
| *Ours ($\beta = 1$)* | **96(97)** | **96(97)** | 72(93) | **57(57)** | **56(57)** | **31(48)** |
| *Ours ($\beta = 1.2$)* | **96(97)** | **96(97)** | 73(**94**) | **57(57)** | **56(57)** | **31(48)** |
| *Ours ($\beta = 1.5$)* | **96(97)** | **96(97)** | 76(**94**) | **57(57)** | **56(57)** | **31(48)** |
| *Ours ($\beta = 2$)* | **97(97)** | **96(97)** | **79(94)** | **57(57)** | **56(58)** | **31(48)** |

Table 6: Fairness metric $\rho$ for all baselines and our method with varying degrees of altruism $\beta$, on MNIST and CIFAR-10 with 5 agents. $\rho$ is computed between **standalone test accuracies** and **final test accuracies**. The higher the values, the better in terms of fairness in rewards.

| Data Partition | MNIST $N = 5$ | | | CIFAR-10 $N = 5$ | | |
|---|---|---|---|---|---|---|
| | UNI | POW | CLA | UNI | POW | CLA |
| *FedAvg* | $-18.6$ | 25.47 | 95.01 | 18.47 | 97.48 | 98.75 |
| q-FFL | 26.46 | 47.26 | 96.07 | 5.53 | 33.25 | 97.60 |
| CFFL | 30.76 | 18.06 | -23.04 | 66.21 | 63.35 | -13.94 |
| *ECI* | 37.18 | 62.13 | 96.41 | 85.43 | 97.86 | 98.45 |
| *DW* | $-33.1$ | 99.35 | 12.11 | 80.64 | 99.17 | 99.90 |
| *RR* | $-47.5$ | 94.84 | 81.36 | 74.43 | $-23.7$ | 97.17 |
| *Ours (EU)* | 71.63 | 70.15 | 91.57 | **96.36** | **99.71** | **99.91** |
| *Ours ($\beta = 1$)* | **83.53** | **99.57** | **98.62** | 85.32 | 95.04 | 99.70 |
| *Ours ($\beta = 1.2$)* | 75.84 | 99.46 | 97.67 | 78.35 | 95.81 | 99.73 |
| *Ours ($\beta = 1.5$)* | 76.92 | **99.57** | 95.37 | 81.05 | 95.56 | 99.72 |
| *Ours ($\beta = 2$)* | 21.16 | -33.99 | 97.72 | 99.22 | 99.89 | 99.97 |

## B.3 Fairness Comparison Based on CGSV

As these FL-based variants all use gradients as the communication medium, we can accordingly adapt our CGSV approximation and the moving averaging formulation as in (4). Specifically, we compute the moving average $r_{i,t}$ for each framework respectively as the (cumulative) contribution of $i$ and use it for fairness evaluation. We calculate the fairness metric $\rho$ by considering two types of reward $\boldsymbol{\xi}$'s: *final test accuracies* in Table 7, and *negative training losses* in Table 8. In comparison, the fairness results in Table 2 are computed between the standalone test accuracies and final test accuracies.

Table 7: Fairness metric $\rho$ for all baselines and our method with varying degrees of altruism $\beta$, on all four datasets. $\rho$ is computed between $(r_{i,T})_{i=1,...,N}$ and **final test accuracies** where $T$ denotes the last iteration as in Table 4. The higher the values, the better in terms of fairness in rewards.

| | MNIST | | | | | | CIFAR-10 | | | MR | SST |
|---|---|---|---|---|---|---|---|---|---|---|---|
| No. Agents | 10 | | | 20 | | | 10 | | | 5 | 5 |
| Data Partition | UNI | POW | CLA | UNI | POW | CLA | UNI | POW | CLA | POW | POW |
| FedAvg | -10.54 | 45.13 | 33.89 | 77.80 | 62.19 | 55.45 | 56.21 | 45.77 | 77.18 | 45.54 | $-7.05$ |
| q-FFL | -86.23 | -15.75 | 67.98 | -54.10 | -35.45 | 12.54 | 61.78 | 36.55 | -58.45 | 49.53 | -93.94 |
| CFFL | 85.89 | 38.64 | 15.80 | -44.25 | 5.62 | -16.38 | 41.76 | -54.63 | 45.32 | 19.00 | 19.45 |
| ECI | 44.17 | 85.74 | 91.15 | 79.03 | 89.69 | 91.81 | 80.67 | 89.57 | **97.30** | 84.58 | 93.12 |
| DW | 0.22 | 88.64 | -40.55 | 65.21 | 91.77 | 73.14 | -10.25 | 87.25 | 9.27 | 73.53 | 72.07 |
| RR | 3.17 | 75.12 | 78.15 | -3.33 | 86.43 | 91.88 | 43.04 | -17.31 | 84.95 | -15.5 | -6.43 |
| Ours (EU) | 49.44 | 79.85 | 83.78 | 55.27 | 91.63 | 85.35 | **89.89** | 93.02 | 87.19 | 87.88 | 93.21 |
| Ours ($\beta = 1$) | **90.49** | 94.68 | 77.27 | **93.20** | 92.89 | **94.58** | 82.66 | 92.75 | 94.89 | 96.01 | 94.31 |
| Ours ($\beta = 1.2$) | 89.74 | **96.28** | 82.95 | 90.65 | **96.99** | 93.42 | 78.99 | **93.41** | 95.96 | **96.12** | 90.00 |
| Ours ($\beta = 1.5$) | 80.23 | 91.40 | **93.60** | 90.89 | 90.31 | 93.66 | 76.42 | 92.89 | 89.52 | 95.32 | 95.11 |
| Ours ($\beta = 2$) | 82.36 | 91.03 | 88.45 | 74.88 | 87.44 | 91.20 | 72.71 | 89.63 | 84.92 | 85.01 | **97.49** |

## B.4 Empirical Validation of Theorem 2 via Test Loss

In addition to the results from Appendix B.3, we perform experiments to further validate Theorem 2 by considering the *test* loss (instead of training loss) as the reward, i.e., a lower test loss corresponds

Table 8: Fairness metric $\rho$ for all baselines and our method with varying degrees of altruism $\beta$, on all four datasets. $\rho$ is computed between $(r_{i,T})_{i=1,\ldots,N}$ and **negative training losses** where $T$ denotes the last iteration as in Table 4. The higher the values, the better in terms of fairness in rewards.

| | MNIST | | | | | | CIFAR-10 | | | MR | SST |
|---|---|---|---|---|---|---|---|---|---|---|---|
| No. Agents | 10 | | | 20 | | | 10 | | | 5 | 5 |
| Data Partition | UNI | POW | CLA | UNI | POW | CLA | UNI | POW | CLA | POW | POW |
| FedAvg | 68.68 | 86.73 | 96.01 | 83.05 | 87.35 | 84.03 | 55.84 | 83.57 | 95.33 | 87.26 | 73.53 |
| q-FFL | 48.04 | 60.71 | 48.68 | 5.78 | 30.44 | -10.48 | 12.65 | 88.00 | -55.00 | **99.39** | 94.39 |
| CFFL | -52.81 | -6.07 | -57.21 | -48.42 | 13.13 | -7.90 | 3.55 | -11.76 | 41.22 | 33.30 | 44.33 |
| ECI | 71.79 | 92.97 | 82.10 | 82.10 | 78.01 | 58.35 | **84.90** | 85.81 | 93.81 | 95.10 | 82.75 |
| DW | 68.12 | **95.13** | 95.59 | 59.05 | 72.13 | 85.23 | 49.31 | 90.17 | 95.36 | 88.87 | 73.12 |
| RR | 46.52 | 87.84 | **96.65** | 31.99 | 92.73 | 91.20 | 16.13 | 85.28 | **97.02** | 87.97 | 73.56 |
| Ours (EU) | 71.59 | 87.32 | 96.11 | 82.14 | 86.08 | 84.00 | 61.53 | 83.31 | 95.12 | 88.73 | 73.23 |
| Ours ($\beta = 1$) | **91.64** | 92.26 | 96.84 | **89.47** | **94.78** | **96.82** | 84.85 | **94.59** | 90.13 | 90.44 | 91.92 |
| Ours ($\beta = 1.2$) | 90.36 | 91.94 | 91.19 | 88.87 | 93.28 | 96.41 | 83.84 | 90.94 | 90.16 | 90.05 | **97.54** |
| Ours ($\beta = 1.5$) | 91.03 | 93.33 | 92.27 | 88.21 | 92.11 | 91.39 | 84.43 | 90.51 | 90.33 | 89.54 | 89.84 |
| Ours ($\beta = 2$) | 85.02 | 88.61 | 94.88 | 87.51 | 90.09 | 92.36 | 78.95 | 88.84 | 90.65 | 88.72 | 94.64 |

to a better reward. Figure 8 demonstrates the same consistent trend as with training losses. This demonstrates the generalizability of Theorem 2 that the agents who upload better gradients have receive better models (i.e., with lower test losses).

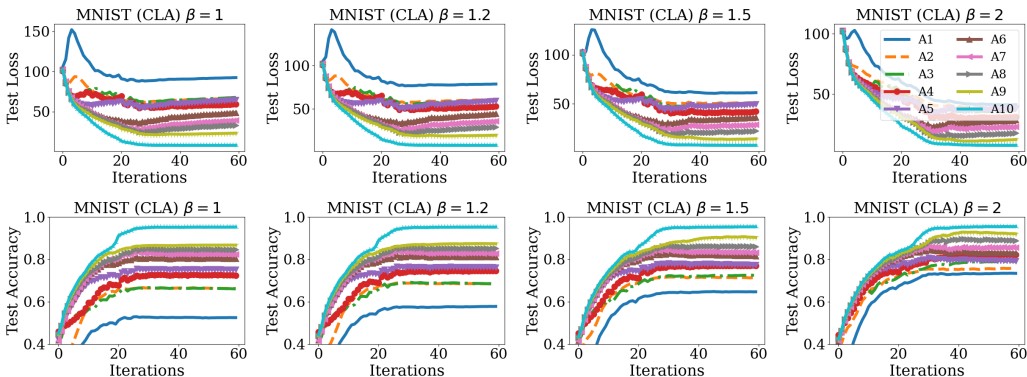

Figure 8: Test *losses* (first row) and test *accuracies* (second row) of the final models vs. the altruism degree $\beta$ for MNIST CLA. From left to right, $\beta = [1.0, 1.2, 1.5, 2]$. A higher $\beta$ leads to better performance for agents with lower contributions.

## B.5 Additional Comparison with q-FFL

Since q-FFL sets out to achieve a different notion of fairness than ours, we perform more in-depth comparison to examine the effects these two algorithms have on the agents' final models. We plot the final performance in terms of test accuracy, test loss and train loss of all 5 agents for MNIST and CIFAR-10 under the three types of data partitions in Figures 9 and 10. And Figures 11 and 12 show the corresponding results for 10 agents.

We observe that in all scenarios, our algorithm performs noticeably better in terms of the final test accuracy. However, it may be due to that in q-FFL each agent is interested in performing well on their *own* local/private test sets (of the same distribution of their local train set). We do not investigate that use case. Instead, our scenario is that all the agent are interested in one common objective (represented by the same test set on which the test accuracy and test loss is computed).

Specifically, we observe from the second row of Figure 9, q-FFL 'under'-optimizes agent 4 while our algorithm fairly evaluates and rewards all the agents (the increasing trend of test accuracy and decreasing trend of train loss). Moreover, we observe from the second row of Figure 10, q-FFL 'equalizes' the performance of the agents in terms of test accuracy and test loss, while our algorithm fairly rewards the agents (the increasing trend of test accuracy and decreasing trend of test and train losses).

In summary, despite the similarity in the keyword terminology, namely fairness, our algorithm is fundamentally different from q-FFL in that in our setting, all agents share one learning objective while in q-FFL each agent has their own learning objective (which may differ considerably from others').

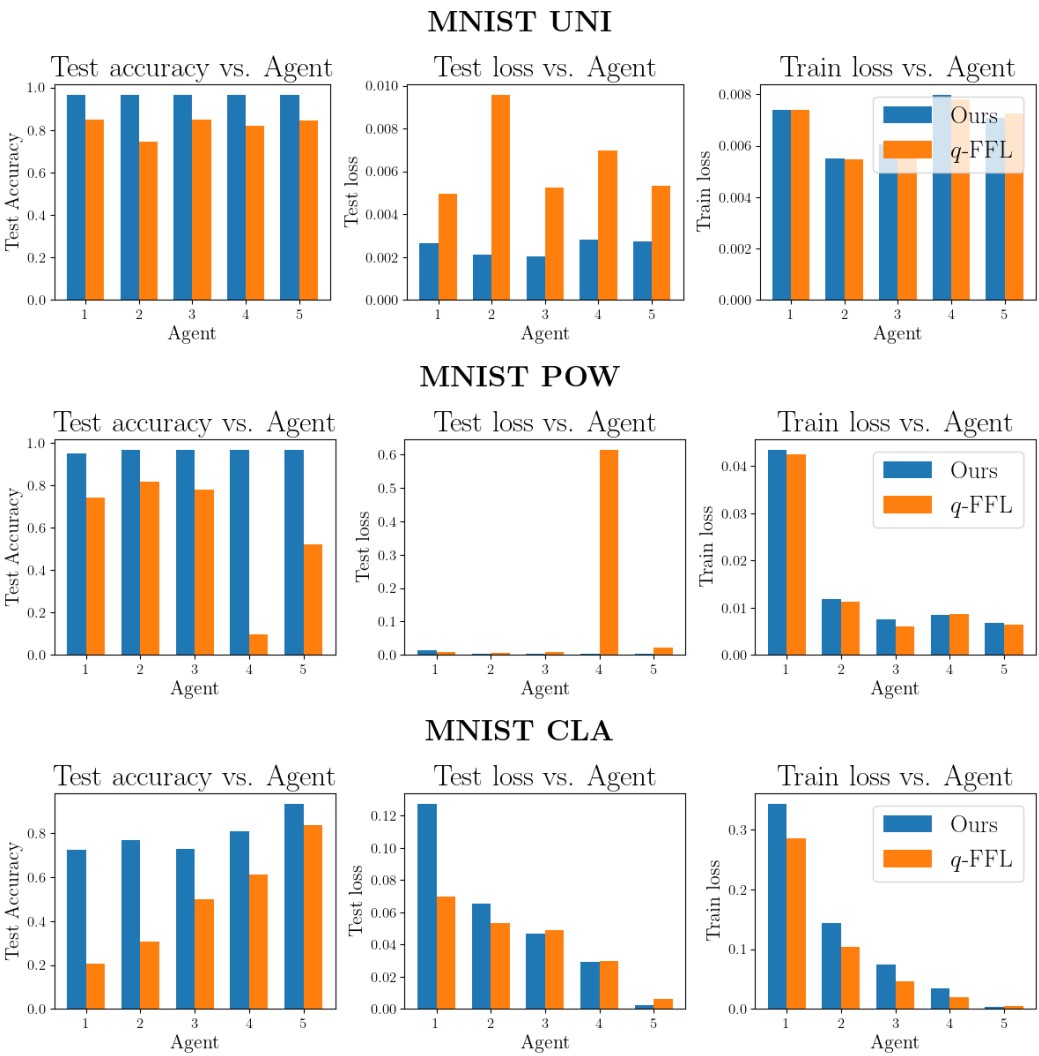

Figure 9: Comparison of final performance across agents between our method and q-FFL.

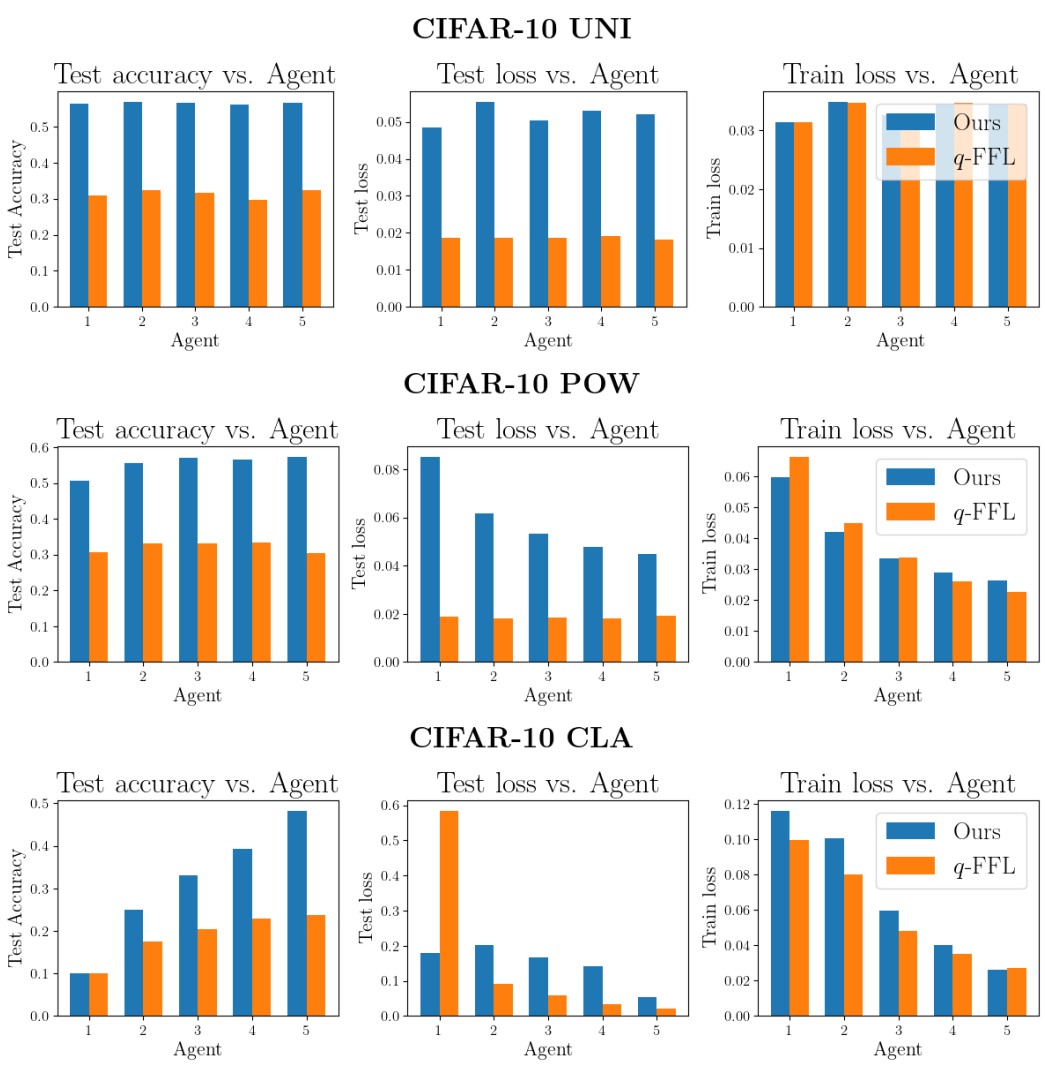

Figure 10: Comparison of final performance across agents between our method and q-FFL.

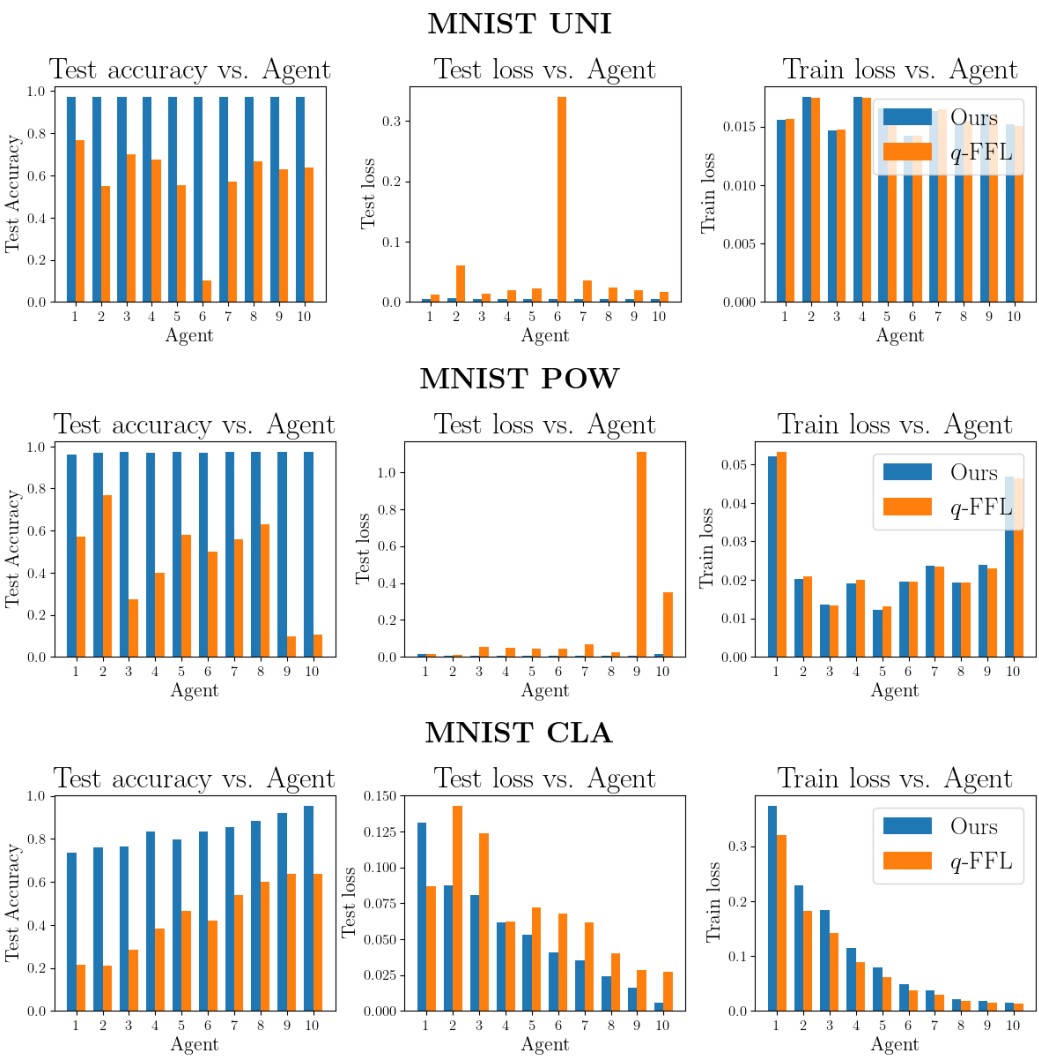

Figure 11: Comparison of final performance across agents between our method and q-FFL.

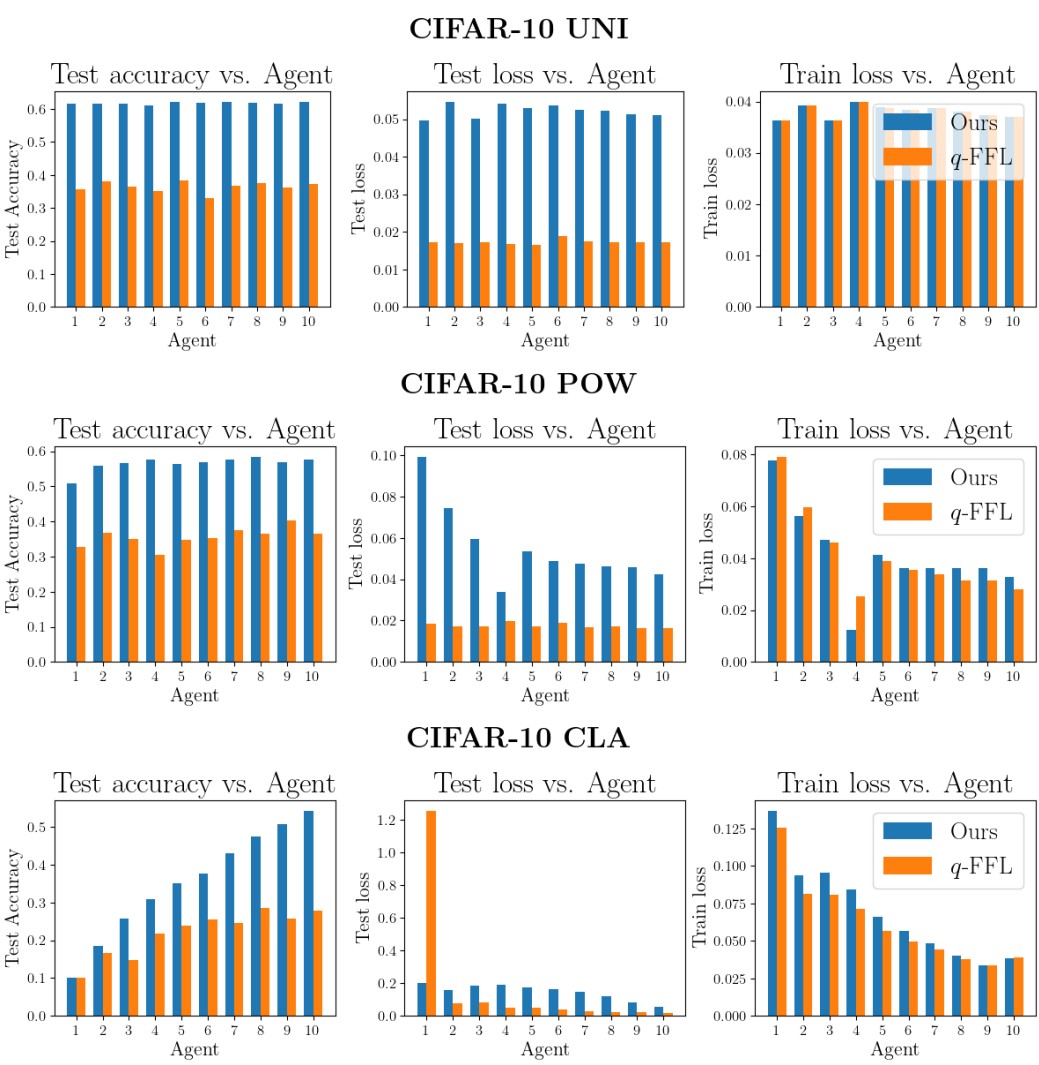

Figure 12: Comparison of final performance across agents between our method and q-FFL.