# OpenReview forum: "Gradient Driven Rewards to Guarantee Fairness in Collaborative Machine Learning"
_NeurIPS.cc/2021/Conference — NeurIPS 2021 Poster_

### Official Review · Reviewer_P3C7 · 2021-07-05

**Rating:** 6
**Confidence:** 4

**Summary:**

This paper proposed an interesting algorithm that aims for collaborative fairness using gradient information (cosine gradient shapley). Overall the paper is well-written and well-motivated. My main concerns are two folded: (1) The current empirical setting is not clear; (2) the fundamental motivation and the potential application of this proposed method is not clear to me. I, therefore, hope authors can elaborate more along those lines.

**Limitations And Societal Impact:**

The defined collaborative fairness somehow conflicts with traditional fairness. I understand that the fundamental goal is different but this proposed algorithm is unfair to minority groups. I, therefore, want to see how to make the proposed algorithm balance between collaborative fairness and traditional fairness (e.g., q-FFL).

**Main Review:**

Pros:

(1) This paper is well-written and well-organized.

(2) The empirical section considers an adequate amount of datasets. In the fairness result, it is interesting to see the proposed algorithm can correct misidentified agent (who was originally mistaken to have low contribution).

(3) The technique/theoretical results are correct to me.

Cons:

(1) The authors define the reward as the quality of the downloaded gradient. Specifically, a local device with a high contribution should be given a gradient that is close to the true averaged gradient while devices with lower contributions should be given sparsified gradients. This strategy intuitively makes sense to me. However, after thinking about it in-depth, I am worried about the convergence behavior. Specifically, since the local gradients are sparsified (contaminated), the trained local model will also be contaminated. Therefore, at the aggregation stage, the aggregated gradient/parameter is problematic. Will this strategy still make the optimizer converges to a stationary point? If not, the authors should clearly discuss this limitation. If yes, what is the intuition behind this? Is there any theoretical justification?

Furthermore, since the aggregated gradient is, in fact, contaminated, it is also unfair to the high-contributed devices: for instance, why should they get this contaminated aggregated gradient/parameter? Under some extreme conditions, why not training solely on their local data/model instead of collaborating with others?

(2) It is not clear to me how to reasonably sparsify the gradient. Suppose all local devices are honest and willing to train a global model. However, some devices have very little data or low-quality data, which they cannot control by themselves. In this case, the proposed algorithm will discriminate against those minorities and assign them low-quality gradient/parameter. This vicious cycle will continue and situations will become worse and worse. Is this still fair?

(3) The experimental setup for results in Table 1 is not clear. Specifically, how many local devices provide low-quality models? If so, what is the reason? Is it because they are unwilling to share gradients, providing fake gradients, or they have low-quality datasets?

(4) Authors mentioned that "we expect the agents with more classes to receive better models". This is fundamentally an unfair argument for fairness, even though the goal is collaborative fairness. Also, this statement clearly assumes all agents are honest or almost honest and classifies agents based on their provided gradients. Under this scenario, what if a local device with a large amount of high-quality data is unwilling to collaborate with others and providing a fake gradient? Will this local device significantly affect other devices and the overall performance?

Overall, if I understand correctly, the proposed model is in fact ranking local devices by the quality of their data/model. This is unfair to honest users with low-quality or small amounts of data. Given this argument, I am unsure why the proposed fairness definition is meaningful. I, therefore, hope the authors can clearly define the real-world justification (not application) scenario of the proposed algorithm. Here kindly note that I am not asking for additional experiments. Instead, I want to see under what scenario will this model be useful and proposed fairness definition is justifiable.

**Time Spent Reviewing:**

4 hours

---

> ### Author Response · Authors · 2021-08-10
> **Author Response**
>
> We thank the reviewer for taking the time to review our paper carefully, pointing out the nice auto-correction behaviour by our algorithm and for verifying our technical/theoretical results.
>
> We would like to address the concerns as follows:
> 1. We wish to clarify that our main result guarantees __the model of a high-contributing agent is closer to the server model__ which is expected to converge [25]. Consequently, the agents’ models enjoy different performance guarantees depending on their contributions and may not converge to the same point. However, importantly, __their final models still outperform standalone models, i.e., they stand to benefit from collaboration__. The intuition for using gradients which may not have been calculated w.r.t the same model can be useful can be understood as follows. [Mandt et al. 2017] describes during an SGD optimization, the model reaches an objective surface and descends to reach the local minimum. In particular, the points on this surface share similar descent directions, i.e., the gradients in our case. Therefore, aggregating them can be useful. Empirically, we observe overall, all agents saw an improvement over their standalone models, including the high-contributing agents. Specifically, in Table.1 both the average and maximum performance across agents saw improvements over the standalone setting. This is true even in highly heterogeneous cases such as CLA where an agent could have all ten classes of MNIST (or CIFAR-10) while the others have only a subset of all classes. In addition to our previous argument, incorporating gradients from other agents __can help with regularization__ and __mitigate overfitting to local data__.
>
> 2. We introduce the altruism degree $\beta$ to address this. Specifically, __increasing $\beta$ (more altruistic) increases the rewards for agents with lower contributions__ (possibly honest agents with lower quality data). Empirically, we demonstrate this via the experimental results in Fig.7. We also provide __an interpretable way to select $\beta$ to provide a quantitative relationship__ of the proportionality between contribution and reward from the perspective of ensuring fairness (lines 229-237).
>
> 3. The __setting on data is described in the row “Data Partition” in Table.1 and detailed in Sec.5.1__. These heterogeneous and qualitatively comparable data partitions are designed to allow us to interpret and evaluate the result of our proposed algorithm. For instance, with all honest agents, the agent with best local data is expected to contribute high-quality gradients in general. Contrast this with a common alternative where each agent has two digits from MNIST [31], it is not immediately clear how to reason about which agent is expected to contribute more. It is thus challenging to evaluate the result for collaborative fairness (in ensuring the rewards are commensurate with the contributions) because it is difficult to reason about the contributions in the first place.
>
> 4. We wish to clarify the __experimental setting of honest agents is for evaluating the results__ and __our algorithm/analysis does not assume honest agents__. In our experiments, we use this setting to reason about the behaviours and provide interpretation of the results. The setting of honest agents and the carefully designed data partitions (qualitatively comparable and interpretable) are to allow us to evaluate the algorithm’s effectiveness in guaranteeing collaborative fairness, not used by the algorithm/analysis per se. For our algorithm, if an agent consistently uploads low-quality gradients (whether due to low-quality dataset, or fake gradients), this agent is expected to receive a commensurate reward. On the other hand, investigating whether and how fake gradients affect the overall performance, can be considered under a separate field of robust learning and not the focus of our work. We address collaborative fairness to ensure the final performance is fair relative to the contributions instead of optimizing robustness because addressing all the non-trivial issues simultaneously would be challenging and difficult to evaluate.
>
> 5. Regarding the definition of collaborative fairness, we wish to highlight this notion of fairness (i.e., the rewards should be commensurate/proportional to the contributions) has its theoretical standing in several fields including __cooperative game theory, mechanism design and computational social choice__ (in lines 19-22) and has seen __applications in machine learning, federated learning and collaborative machine learning__ (in lines 26-28). Furthermore, this notion of fairness is adopted in designing suitable rewards to FL/collaborative ML agents to incentivize them to contribute (in lines 34-35, 72-73). It is often formalized via the Shapley value, so we adopt it to define our Cosine gradient Shapley to leverage the available gradients in the setting. Table 2 (in main text) shows our method is more effective in guaranteeing our notion of fairness as in Definition 3.
>
>     We provide a more specific justification and more references as requested by the reviewer. [46] detailed one such justification: collaboration among financial institutions (involving and raised by WeBank). To illustrate, these banks are willing to collaborate if their respective final payoffs are commensurate with their contributions. It is crucial to them because giving equal payoffs to their collaborators (i.e., other banks, who are also competitors for the same market share) who contribute considerably less would highly disincentivize this form of collaboration. Our proposed definition of fairness is of particular use to address this challenge, which is also observed in medicine [Weinstein et al. 2011, Drazen et al. 2016], real estate [Conway. 2018] and agriculture [Claver. 2019].
>
> 6. Regarding comparison with q-FFL, we wish to point out __these two notions are not necessarily conflicting__ as we empirically demonstrate. First, we provide our explanation. Our method and q-FFL target different types of performance. __Our method focuses on the final test performance__ (on a hold-out test set statistically similar to the true data distribution) while __q-FFL focuses on all local training losses__ on local/personalized datasets (which could be statistically different from the true data distribution). For instance, the true data distribution for MNIST is assumed to have equal support on all ten digits (as is the hold-out test set we use) while the local datasets highly depend on each agent and may contain only a subset of digits (e.g., in CLA or [31]). For our initial remark, we illustrate with the highly heterogeneous data partition CLA: Agent 1 has only one digit locally while Agent 5 has ten digits locally. For both to achieve equitably low training losses (q-FFL), it is expected that Agent 5 has better performance on the hold-out test set that contains all ten digits (our method). Notably, an important assumption of q-FFL is that it explicitly depends on the true local training losses reported by the agents. Dishonest agents could exploit this by over-reporting local training losses to receive more “attention” in optimization. In contrast, our method does not require this assumption as the server evaluates their contributions based on uploaded gradients so uploading low-quality fake gradients will decrease their measured contributions and rewards.
>
>     Empirically, our additional comparison with q-FFL (on MNIST and CIFAR-10 under all three data partitions) in Appendix B.5 examines the performance distribution among the agents. We consider final test performance (loss & accuracy) and local training loss. q-FFL claims to equalize local training losses (shown in the 3rd column of Figs.9-12), __the comparison shows our method and q-FFL provide highly similar results w.r.t local training losses__. On the other hand, for final test performance, results for our method in 1st & 2nd columns show the trend that higher contributing agents receive better test performance as compared to other agents, confirming our fairness results in Table.1. Consequently, __our method achieves our notion of collaborative fairness, and empirically performs similarly to q-FFL w.r.t their equality fairness__.
>
> We hope your clarifications could improve your opinion of our paper.
>
> #### References:
> Mandt, S., Hof Fman, M. D., & Blei, D. M. (2017). Stochastic gradient descent as approximate Bayesian inference. Journal of Machine Learning Research, 18, 1–35.
>
> Claver, H. Data sharing key for AI in agriculture. Future Farming, Feb. 2019. URL https://www.futurefarming.com/Tools-data/Articles/2019/2/Data-sharing-key-for-AI-in-agriculture-389844E/.
>
> Conway, J. Artificial Intelligence and Machine Learning: Current Applications in Real Estate. PhD thesis, Massachusetts Institute of Technology, 2018.
>
> Weinstein, M., & Skinner, J. (2011). Data Sharing from Clinical Trials — A Research Funder’s Perspective. New England Journal of Medicine, 362(5), 567–571.
>
> Drazen, J. M., Morrissey, S., Malina, D., Hamel, M. B., & Campion, E. W. (2016). The importance - And the complexities - Of data sharing. New England Journal of Medicine, 375(12), 1182–1183.

---

> > ### Comment · Reviewer_P3C7 · 2021-08-10
> > **New comments**
> >
> > Dear Authors,
> >
> > Thanks for your detailed reply. After reading your rebuttal, I have further questions. I will use the index number provided in your response:
> >
> > 1. This part is still unclear to me. Suppose loss functions are convex. There are 3 agents and $\bar w^{(t)}=0.4 w^{(t)}_1+0.3w^{(t)}_2+0.3w^{(t)}_3$, where $\bar w, w$ are model parameters and $(t)$ is the current iteration (suppose an aggregation step happens at this iteration). In the conventional FL setting, it is known that $w^{(t)}\to w^*$ as $t\to\infty$ no matter data partitions are iid or non-iid. Now, in your scenario, suppose agent 3 did a bad contribution and some of his gradients are zero-out. In this case, agents 3 will get an inaccurate aggregated gradient. Then agent 3 will update its model parameter based on this inaccurate gradient. In the next aggregation step, the server will again aggregate models using this inaccurate gradient. Despite the gradients from agent1/2 are, say, 100% correct, the impact of agent 3 is large. Therefore, I am note sure why the aggregated model parameter will finally converge. Furthermore, since gradients from some agents are spasified and aggregation will certainly takes those spasified agents into consideration, it is unclear why the final model outperform standalone model. In particular, a agent will high-contribution will definitely be affected by other low-quality models.
> >
> > You mentioned that"the model of a high-contributing agent is closer to the server model which is expected to converge". This is only true when aggregation does not happen. Overall, my concern is that the convergence behavior does not make sense intuitively. It would be very helpful if the authors can provide some theoretical analysis.
> >
> > 2. This makes sense. Thanks for your clarification. My question is: suppose $\beta$ is very large (the reward to agent is same or extremely close to true gradient), will the proposed model become FedAvg but with different weight parameter $p_i$?
> >
> > 4. I agree that this can be a further research direction. Currently, I won't require any further analysis since it looks demanding.
> >
> > Overall I think my concerns 2-6 have been well-addressed. I will increase my rating to 5. However, I am still hesitant to recommend "accept" due to concern 1. Specifically, the convergence behavior does not make sense both intuitively and theoretically. I hope the authors can make more clarification.

---

> > > ### Author Response · Authors · 2021-08-11
> > > **Author response to new comments**
> > >
> > > We thank the reviewer for the swift and detailed response. We would like to address the comments as follows,
> > >
> > > 1. We wish to clarify __uploaded gradients from agents are not sparsified, only normalized__ with $\Gamma$, only the downloaded gradients by the agents are sparsified. We then describe an _informal_ analysis for the purpose of providing some intuition.
> > >
> > >     - Following your example, let $u_{i}^{(t)}$ denote the uploaded gradient (normalized) from $i$ in iteration $(t)$ and abuse the notation a bit to denote $u_{1,2}^{(t)} = r_1^{(t-1)} \times u_1^{(t)} + r_2^{ (t-1)}\times u_2^{(t)}$ similar to our aggregation rule in Equ.(1). For simplicity, assume agents 1 & 2 have always received the full gradient so their respective models are the same as the server model.
> > >
> > >
> > >         Agent 3 downloads a sparsified gradient which makes $w_3^{(t+1)}$ not as good as $w_1^{(t+1)}$ or $w_2^{(t+1)}$ by design. For this analysis, we adopt an alternative perspective that $w_3^{(t+1)}$ is _stale_, in the sense that some of components of the aggregate gradient are 'delayed indefinitely' to be applied to $w_3^{(t+1)}$ [2]. $u_3^{(t+1)}$ is hence a stale gradient. But it is still useful if it aligns in its direction with the non-stale gradient (i.e., gradients calculated from the most updated model). [10] defined a _gradient coherence_ to characterisze this 'usefulness' with the cosine similarity between a non-stale and a stale gradient. Here, the non-stale gradient is $u_{1,2}^{(t+1)}$ and the stale gradient is $u_3^{(t+1)}$. [10] provides a convergence result for the server model (which could receive both stale and non-stale gradients asynchronously). Specifically, the server model converges to a stationary point with rate $\mathcal{O}(1/\sqrt{T})$ where $T$ is the total number of iterations, under the condition that the gradient coherence is mostly positive, i.e., _for most iterations_, either the server gets updated with the non-stale gradient, or gets updated with a stale gradient that is aligned with the non-stale gradient in the sense of cosine similarity.
> > >
> > >         Our framework is synchronous in implementation and communication. However, it can be seen as an implicitly asynchronous process involving stale and non-stale gradients (due to sparisifcation), w.r.t the gradients the server model receives. Kindly note the above explanation is not a rigorous argument (which would need additional non-trivial formalisation). We hope it can provide some intuition for analyzing the convergence of the server model.
> > >
> > >     - The aggregation of uploaded gradients (which are not sparsified and only normalized) in a sense aggergates the useful information from all agents, and __could improve over the standalone performance__. A first possible reason is in the aggregation each uploaded gradient is multiplied by a weight $r_i^{(t)}\leq 1$, so the __variance in the noise of each gradient is reduced in a quadratic manner__. Secondly, we can find __a middle-point__ (by varying $\beta$ and sparsification) __to balance performance improvement and collaborative fairness__. In the extreme case where all local models are completely synchronized, it is expected that training together should improve standalone performance even for a high-contributing agent by learning generalizable information across agents. As the difference in the local models increases (due to sparsification), the benefit of training together diminishes gradually. Consequently, there should exist a middle point (between completely same models and very different models from high sparsification) such that training together is still beneficial but the local models are sufficiently different that they satisfy our notion of collaborative fairness. Empirically, we observe this middle point can be controlled by $\beta$, i.e., larger $\beta$ means smaller difference among models and vice versa.
> > >
> > >
> > > 2. Note  $\beta$ affects the gradients the agents download, while __the aggregation based on__ $r_{i, (t)}$ __is not directly affected by__ $\beta$. Suppose $\beta$ is large enough that each agent downloads the full gradient without sparsification, it is still _not_ FedAvg because the values $r_{i, (t)}$ are being dynamically updated as the training proceeds where as the weight parameter $p_i$ in FedAvg is fixed according to the size of the local data. In particular, $r_{i, (t)}$ may not follow the size of local data. In the example that you have pointed out where "the proposed algorithm can correct misidentified agent" (via observing the values of $r_{i, (t)}$), it shows an additional use for $r_{i, (t)}$ instead of only the weight parameters to aggregate the gradients. In this example (Fig.4 bottom): The data partition is CLA, i.e., the agents' local data differ in the number of classes but have the same size. For FedAvg, the $p_i$ would have been the same for all. Our alrgoithm updates $r_{i, (t)}$ to be higher for agents with more classes of data in their local data.

---

> > > > ### Comment · Reviewer_P3C7 · 2021-08-25
> > > > **Update**
> > > >
> > > > Thanks for the reply. I think you did a good explanation for part 1. Here it would very helpful and insightful to add theoretical convergece analysis to this paper (or to a follow-up work). Specifically, will the sparsified gradient cause slower convergence? Rather than giving a big O expression, it is interesting to see all constant terms explicitly. I understand that it is burdensome to do so during the rebuttal period so I will not ask for any additional update.
> > > >
> > > > Overall I think this is an interesting paper and I am bound to vote for weak acceptance.

---

> > > > > ### Author Response · Authors · 2021-08-27
> > > > > **Thanks to Reviewer P3C7**
> > > > >
> > > > > We would like to thank the reviewer for taking the time to review our paper and the detailed feedback, and in particular for recognising _the technical/theoretical results_ and for the follow-up clarifying questions. We truly appreciate the reviewer's _valuable and positive feedback_.

---

> > > > > ### Author Response · Authors · 2021-08-29
> > > > > **Gentle reminder**
> > > > >
> > > > > We truly appreciate the reviewer's vote of confidence and wish to gently remind the reviewer to update the latest rating.

---

### Official Review · Reviewer_USs6 · 2021-07-15

**Rating:** 5
**Confidence:** 4

**Summary:**

This paper explores the problem of unequal contribution in a gradient-based collaborative learning. The authors propose Cosine Gradient Shapley (CGS) measure together with gradient compatibility function to evaluate shared gradient qualities to assign a fair reward to contributing agents. Extensive set of experiments were conducted to support the proposed ideas and show effectiveness of the approach.

**Limitations And Societal Impact:**

Authors noted potential limitations in their work in section 4.

**Main Review:**

This work builds upon existing solutions by improving the reward allocation function through a CGS. One of the main advantages of the work is that authors provide an approximation of CGS, since an original measure is computationally expensive. This submission is technically sound and properly supports claims by theoretical analysis.

While most of the paper is clear, I would recommend making a number of corrections. First, “Fairness” is a separate direction of research in ML. It is clear in what context authors are using term “fairness”, I would still recommend changing it and use something like “fair distribution”, i.e. clarify context of fairness.

Please clarify the following question: since it was not clear to me after reading the paper. For those agents, who fail contributing good gradient since the start of the training, is there a chance for them to recover? Or they end up crossing the point of no return? The second question is: what would be edge cases in your solution? And how to properly address them? For example, imagine the situation that in the beginning of the training all returned gradients are pointing into different directions equally distant from each other. What to do? And is this situation possible? Or what if we have agents, which dominate the training and getting most of rewards, are we missing valuable gradient corrections from other agents in this case? Can we check it somehow?

Some comments about proofreading and typos, e.g. line 81 “ the following the optimization” and line 179 “leads to to …”

The information on experiments and hyperparameters is enough to reproduce the results. Here are some suggestions on the Experiment section: Pictures are extremely small. While I can zoom in on them on my PC, it is impossible to read them on a printed version. How are hyperparameters picked? e.g. degree of altruism, what works for which tasks.

Related Work section needs to be improved. Please elaborate more on the existing reward designs and combine section “Reward designs in CML” and “Reward selection” and provide more discussion on advantages and disadvantages of existing works, e.g. in abstract authors mention a necessity of using a validation dataset.

**Time Spent Reviewing:**

3

---

> ### Author Response · Authors · 2021-08-10
> **Author Response**
>
> We thank the reviewer for taking the time to review our paper carefully pointing out the typos and the suggestion to improve clarity. We would like to address the concerns as follows:
> - We will carefully modify the phrasing to be more suitable and clearer in our context.
> - We wish to address your questions separately for clarity.
>     - __Figure.4 (bottom) demonstrates such a case of recovery.__ Under the CLA partition for CIFAR-10. The 3rd agent (brown line) with the highest quality and diverse dataset was initially deemed to be of low contribution but our algorithm quickly corrected it.
>     - We wish to highlight the proposed cumulative update of $r_{i,(t)}$ (with a decaying coefficient) can effectively address this so __the effect due to noise from the beginning diminishes over time__. The scenario where uploaded gradients “counteract” each other (e.g., in opposite directions due to noise in training) and lead to failure to generalization is __commonly observed in federated learning__ [5,6,7] and __not unique to our algorithm__. We wish to clarify tackling it is not our main contribution and if remedies are proposed to adequately address it, our method can also benefit. In our method, if such edge cases occur infrequently, our approach of a cumulative $r_{i, (t)}$ can mitigate it to some degree as the effect of edge-cases is comparatively less significant.
>     - The __dominating scenario is rare__ and __occurs under highly heterogeneous data partitions__, and we can check for it.  In our experiments, we verify the agent with low $r_{i,(t)}$ value indeed has extremely low contributions, and we are not missing out on valuable gradient information. We first describe why its occurrence is highly unlikely and then specifically analyse the case in our experiment.
>
>         In the aggregate gradient, each individual gradient is weighted by their respective $r_{i,(t)}$, as in Equ.(1) in line 86. Further, __each uploaded gradient is normalized with the same__ $\Gamma$. Subsequently, the $r_{i,(t)}$ values are cumulatively maintained which means its previous values can provide a stabilizing effect and prevent them from increasing/decreasing indefinitely. During the execution, the server can explicitly check $r_{i,(t)}$ to identify dominating agents. For instance, Fig.4 plots the values of $r_{i,(t)}$ for the 1st and 3rd agent under three separate data partitions. We also observe in our most experiments $r_{i,(t)}$ tend to stabilize over iterations to values that are not extremely large Fig.4 (bottom). In contrast, we observe an extreme case (Fig.4 top) where the 3rd agent under CLA has a large $r_{i,(t)}$. Recall under this CLA data partition, Agent 1 has only one class of data out of ten classes, and Agent 3 has all ten classes in local data. In less extreme data partitions (in terms of heterogeneity), we rarely observe the dominating values of $r_{i,(t)}$. Under the same experiment, Fig.5 (left) shows the l2 difference between the downloaded gradient (reward) and the aggregated gradient, so a lower l2 difference means a higher reward. The purple dashed line is Agent 1. We clearly see that despite a very low $r_{i,(t)}$ for Agent 1 is low, the l2 difference is actually still quite low (the quality of the reward is still reasonable). To understand this, we highlight that our reward scheme is non-rivalrous, meaning one agent getting a high reward does not automatically reduce the reward available for the remaining agents. We introduce the altruism degree $\beta$ to flexibly control the proportionality between rewards and contributions. Following our description in lines 229-237, we give a simple example as follows, suppose $\beta=2$, and $r_{i,(t)} =0.2$ and $\max_{j}r_{j,t}=0.25$, then the sparsification degree for Agent i is $\leq 17\\%$. This indicates that while Agent i contributes approximately $80\\%$ w.r.t highest contributing agent, the reward for Agent i is more than $83\\%$ of what the highest contributing agent receives. In addition, the sparsification starts from the values of the smallest magnitude and preserves the gradient as much as possible. So intuitively, while the agents with lower $r_{i,(t)}$ receive lower rewards (as desired), their rewards are nonetheless quite good, hence the result in Fig.5 left for Agent 1.
>
>         Furthermore, we wish to clarify that in this extreme case why we are not missing out on valuable gradient information from Agent 1. Recall Agent 1 has only one class of data, while Agent 3 has all ten. If the uploaded gradient from Agent 1 has a relatively large weight, the model would tend to overfit to the one class of data, which could deteriorate the performance. On the other hand, the information contained in Agent 1’s gradients is also contained in both Agents 2 & 3’s, hence we are not missing out on valuable information from Agent 1.
>
> - Thank you for pointing out typos. We will correct them.
> - Thank you for the suggestion, we will adjust the layout to allow larger figures and diagrams for easier viewing.
> - We describe __an interpretable way to select $\beta$ to provide a quantitative relationship__ of the proportionality between the contribution and reward of ensuring fairness (lines 229-237). Relating to learning performance, as $\beta$ affects the sparsification degree, it is also model and task dependent. Specifically, we want to set a $\beta$ such that it achieves desired fairness without compromising learning performance. Notably, __setting a sparsification degree optimal for learning is not unique to our setting__. [2,3] provide empirical evidence that the best sparsification degree (w.r.t learning) can vary significantly depending on the data/model/learning task. [4] provides a theoretical result for convex and smooth objective functions, but the expected convergence depends on the sparsification degree in a complicated way via big-Oh notations, so it is not obvious how best to set it.
>
>     Consequently, we follow our described way to experiment with several such $\beta$ values and report all the corresponding results as in Tables.1 & 2 and Figs.3 & 7. Our results suggest that generally a higher $\beta$ (so a lower sparsification degree in general) would give better performance, which is intuitive since overall the agents download better gradients. The other hyperparameters: $\Gamma$ (gradient normalization) is empirically found so that gradient explosion does not occur; for $\alpha$, we conduct grid-search to find the best for the model/task combination.
>
> - Thank you for your suggestion, we will improve our discussion on reward designs to include a better-rounded comparison with existing works to highlight the advantages and disadvantages.
>
> We hope the above clarifications could improve your opinion of our paper.
>
> #### References:
> [1] Mandt, S., Hof Fman, M. D., & Blei, D. M. (2017). Stochastic gradient descent as approximate Bayesian inference. Journal of Machine Learning Research, 18, 1–35.
>
> [2] Yan, Z., Xiao, D., Chen, M., Zhou, J., & Wu, W. (2020). Dual-Way Gradient Sparsification for Asynchronous Distributed Deep Learning. ACM International Conference Proceeding Series.
>
> [3] Aji, A. F., & Heafield, K. (2017). Sparse communication for distributed gradient descent. Proc. EMNLP, 2015, 440–445.
>
> [4] Stich, S. U., Cordonnier, J. B., & Jaggi, M. (2018). Sparsified SGD with memory. Advances in Neural Information Processing Systems.
>
> [5] Malinovskiy, G., Kovalev, D., Gasanov, E., Condat, L., and Richtarik, P. From local SGD to local fixed-point methods for federated learning. In Proceedings of the 37th International Conference on Machine Learning, 2020.
>
> [6] Pathak, R. and Wainwright, M. J. FedSplit: An algorithmic framework for fast federated optimization. In Advances in Neural Information Processing Systems, pp. 7057–7066, 2020.
>
> [7] Charles, Z. and Konecný, J. Convergence and accuracy trade-offs in federated learning and meta-learning. In Proceedings of The 24th International Conference on Artificial Intelligence and Statistics, 2021.

---

> ### Author Response · Authors · 2021-08-25
> **Thanks to Reviewer USs6**
>
> We would like to thank the reviewer for taking the time to review our paper and the valuable feedback, and in particular for verifying the _theoretical analysis_ and for recognising the _technical soundness_ of our work.
>
> We hope our response pointing to the relevant empirical results and analysis were adequate in answering your questions regarding the behaviour of agents and whether there would be dominating edge cases. Please let us know if you have further clarifying quetsions. We truly appreciate your valuable and detailed comments about clarification of the context of ''fairness'' and the writing suggestions on related work and increasing the figure sizes for better viewing.

---

> ### Author Response · Authors · 2021-09-02
> **Gentle reminder for response**
>
> We would like to gently remind the reviewer of any follow-up clarifications or questions that we can do our best to address in the remaining time. We hope our previous response has clarified your comments on evaluating the quality of the uploaded/contributed gradients and possible edge cases (and how to check \& address them). We hope it has helped improved your opinion of our work. We truly appreciate your comments on re-organisation of related work and increasing figures for improved viewing and will incorporate them in our revised version. Please let us know if there are additional comments you have for us.

---

### Official Review · Reviewer_Frjv · 2021-07-16

**Rating:** 6
**Confidence:** 3

**Summary:**

The paper studies a collaborative setting where platform aims to facilitate self-interested agents pooling their resources towards a common learning task. Each agent calculates the gradient on their local objective that is based on their local dataset, and the common goal is to optimize a weighted sum of these objectives. The author propose that agents be given sparse approximations of the average gradient, where the approximation quality matches the ranked order given by the Shapley value, where the value of a subset is the angle between the average gradient reported by agents on that subset and the average gradient across all agents. They prove that agents who upload better agents have better predictive performance guarantees and evaluate their approach on several datasets.

**Limitations And Societal Impact:**

The authors adequately discuss the societal impact within the paper.

**Main Review:**

While collaborative learning is becoming more mainstream in applications, there has been limited attention in the academic literature on how to best facilitate the pooling of resources by self-interested agents. The idea of using the Shapley value (which is standard in cooperative game theory) to determine rewards in the context of collaborative learning is very natural and interesting. The approach of giving agents information whose “quality” is based on the Shapley value, rather than simply using the Shapley value to determine monetary transfers, is very interesting.  The paper is generally well-written.

One weakness is with the definition of Cosine Gradient Shapley Value. It is not clear that taking the inner product with the average gradient reported by all agents is always meaningful. Even if the agents start at the same point at initialization, their trajectories might diverge significantly since they are given different gradients, and so the geometric proximity to the average gradient may not be an appropriate measure of quality.

Another weakness is that the authors do not provide theoretical insights into how the predictive performance for each agent changes with beta (the “altruism degree”) or how the rate of convergence changes with beta.

Lastly, regarding the estimator of Cosine Gradient Shapley Value in (3), this estimator does not seem to be unbiased, and the resulting bound in Theorem 1 requires an assumption on the inner products.  The empirical comparison against typical sampling-based approaches is fairly limited, and it would be helpful to discuss the advantages and disadvantages of the proposed estimator.

It would also be helpful to discuss motivating applications where agents give gradients at each step, rather than directly sharing their data or their learned model with the platform.

----------
Update: Thanks to the authors for their detailed responses! I appreciate the authors' additional justifications of Cosine similarity (e.g. in relation to the cited work below and also in Appendix A.1), and have raised my score.


**Time Spent Reviewing:**

4

---

> ### Author Response · Authors · 2021-08-10
> **Author Response**
>
> We thank the reviewer for taking the time to review our paper and for finding our approach of leveraging the Shapley value in determining the reward interesting. We would like to address the concerns as follows:
>
> - __Comparing gradients with cosine similarity is useful to examine their qualities__ (even stale ones calculated w.r.t older, slightly different models) [1,2], with the assumption these different models are close. To ensure this, all agents are __initialized with the same model__; and their __downloaded gradients are as close to the aggregated gradient as possible__. Specifically, the sparsification (to construct the downloaded gradient) starts from the values of the smallest magnitude and the sparsification degree can be flexibly set according to the altruism degree $\beta$. For instance, suppose $r_{i,(t)}=0.2$ and $\max_{j}r_{j,t}=0.25$, then the sparsification degree is $\leq 17\\%$. Empirical evidence in Fig.5 also demonstrates that the downloaded gradients among agents have relatively small $l_2$ difference, except for cases the agents data are of considerably lower-quality such as in CLA. Furthermore, Fig.6 shows that the $l_2$ norm of the model difference between an agent and the server model in the last layer is also relatively small.
>     - Furthermore, we relate to an observed phenomenon in SGD regarding the gradients as noted in [3] to justify our use of cosine similarity. If the models are close (on the same objective function surface), then their calculated gradients (even if the models are different) point to similar descent directions, which can be quantitatively measured by cosine similarity.
>
> - We introduced the altruism degree $\beta$ to flexibly control how ‘strictly’ the rewards follow the Shapley value, instead of improving convergence. Intuitively, the __altruism degree determines how closely the local models follow the server model, not how fast local/server models converge__. This means we can expect the local models to perform better with larger $\beta$, not necessarily to converge faster. We also provide an interpretable way to quantify this relationship in lines 229-237. Furthermore, empirical evidence (Fig.7 in main text & Fig.8 in Appendix B.3) demonstrates the desired effect that increasing $\beta$ (being more altruistic) allows more agents, especially the ones with lower quality data, to have better performance.
>
> - Regarding our proposed estimator, we discuss the assumptions required to have small errors in lines 137-139 and interpret their implications, including the lower bound of the dot product. Further discussion can be found in lines 531- 535 in Appendix A.1. Empirically, __our estimator outperforms the unbiased Monte-Carlo (MC) estimator, in terms of estimation error and runtime efficiency__. Fig.1 (green lines) shows the preliminary comparison result. Also we wish to clarify that this comparison is limited due to high computational cost of exact calculation and also the MC estimator which scales poorly. In the experimental section, we implement MC estimator for ECI (with a more relaxed error requirement than in the preliminary comparison) and the results in Table.3 demonstrate its runtime inefficiency. Results in Tables.1 & 2 demonstrate our method outperforms ECI, in both fairness and learning performance. We attribute this to the suitability of the proposed estimator with the specific problem/setting, and its capability in effectively leveraging the assumptions which are empirically observed. In contrast, the simple alternative of Euclidean distance under-performs our estimator considerably in terms of both fairness and learning performance.
>
> - A practical motivation is __better preservation of private data__ which are kept locally throughout, a commonly adopted reason for federated learning and collaborative machine learning. Furthermore, __gradients may be more useful and easier to evaluate than models__. Gradients which carry information about the descent direction can be useful even if they are not calculated from the most updated models [1]. Evaluating gradients using our method circumvents the need for a validation set to determine the model performance. __Using gradients offers flexibility to differential-private techniques__ such as differentially-private SGD by each individual agent without needing to change the algorithm at the server side. We will include in our discussion to further highlight the advantages of using gradients.
>
> We hope the above clarifications could improve your opinion of our paper.
>
> #### References:
> [1] Dai, W., Zhou, Y., Dong, N., Zhang, H., & Eric P. Xing. (2019). Toward Understanding the Impact of Staleness in Distributed Machine Learning. Proc. ICLR.
>
> [2] Chen, M., Mao, B., & Ma, T. (2021). FedSA: A staleness-aware asynchronous Federated Learning algorithm with non-IID data. Future Generation Computer Systems, 120, 1–12.
>
> [3] Mandt, S., Hof Fman, M. D., & Blei, D. M. (2017). Stochastic gradient descent as approximate Bayesian inference. Journal of Machine Learning Research, 18, 1–35.

---

> ### Author Response · Authors · 2021-08-25
> **Thanks to Reviewer Frjv**
>
> We would like to thank the reviewer for taking the time to review our paper and the valuable feedback, and in particular for recognising the _interesting aspect of our proposed utilisation of the Shapley values_ to design the rewards.
>
> We hope our response has adequately addressed your comments regarding our approach of comparing gradients via an inner product, the effects of $\beta$ and the analysis of our proposed estimator. Kindly let us know if anything is unclear. In addition, we hope we have clarified the motivating applications of sharing gradients instead of models or data. We truly appreciate your valuable feedback and comments that help us further highlight/clarify the important parts of our work.

---

### Official Review · Reviewer_ztAX · 2021-07-19

**Rating:** 7
**Confidence:** 4

**Summary:**

A gradient-based solution to the fair reward allocation problem in gradient-based collaborative ML/FL with theoretical guarantees on fairness. The authors proposed a Cosine Gradient Shapley to formalize the fairness.


**Limitations And Societal Impact:**

Yes

**Main Review:**

Strengths:
1. An efficient approximation to the Cosine Gradient Shapley to address the computational issue of the Shapley value.
2. No validation dataset is required to evaluate participant's contribution.
3. Extensive experiments to demonstrate the effectiveness of the proposed solution in achieving fairness.

Weakness:
1. The high-level explanation on how fairness is achieved using sparsification of gradients can be clearer.
2. More comparison with existing methods such q-FFL should be added.

**Time Spent Reviewing:**

5

---

> ### Author Response · Authors · 2021-08-10
> **Author Response**
>
> We thank the reviewer for taking the time to review our paper. We would like to address the concerns as follows:
> 1. In our notion of collaborative fairness, the __agents who contribute better gradients are to be rewarded better, with better gradients during download__ (resulting in better final models). To achieve this, we control the quality of gradients the agents download via sparsification, namely, a higher degree of sparsification would correspond to a lower quality gradient after sparsification. Consequently, for agents with a better contribution record ($r_{i,(t)}$), they correspondingly download a less sparsified gradient. Theoretically, we ensure that agents with higher ($r_{i,(t)}$) enjoy better model convergence guarantees; while empirically we also demonstrate this.
>
> 2. Our method tries to achieve that the agents receive rewards (their final test accuracy) commensurate with their contributions (estimated by their standalone test accuracies or $r_{i,(t)}$). q-FFL tries to equalize the local training losses among all the agents. __Fundamentally these two methods take different perspectives of “fairness”__. In particular, our notion of fairness originates from cooperative game theory, mechanism design, and computational social choice (in lines 19-22), and has seen applications in machine learning and federated learning (in lines 26-28). It is often formalized via the Shapley value, so we adopt it to define our Cosine gradient Shapley to leverage the gradients in collaborative machine learning. Table 2 (in main text) shows our method is more effective in ensuring our notion of fairness as in Definition 3.
>
>     Additional empirical comparison with q-FFL (on MNIST and CIFAR-10 under all three data partitions) is included in Appendix B.5 to examine the performance distribution among the agents. We consider final test performance (loss & accuracy) and local training loss. q-FFL claims to equalize local training losses (shown in the 3rd column of Figs.9-12), __the comparison shows our method and q-FFL provide highly similar results w.r.t local training losses__. On the other hand, for final test performance, results for our method in 1st & 2nd columns show the trend that higher contributing agents receive better test performance as compared to other agents, confirming our fairness results in Table.1. Consequently, __our method achieves our notion of collaborative fairness, and empirically performs similarly to q-FFL w.r.t their equality fairness__.
>
>
> We hope the above clarifications could improve your opinion of our work.

---

> > ### Comment · Reviewer_ztAX · 2021-09-03
> > **Response**
> >
> > Thanks for clarifying the intuition and adding discussion about the comparison method, and it is also helpful to incorporate such explanations or discussion in the new version of the paper. I think this paper can be interesting, and will raise the score to 7.

---

> ### Author Response · Authors · 2021-08-25
> **Thanks to Reviewer ztAX**
>
> We would like to thank the reviewer for taking the time to review our paper and the valuable feedback, and in particular for recognizing the strengths of our in terms of _theoretical analysis_ and _empirical experiments_.
>
> Kindly let us know whether we have adequately addressed your comments on improving the clarity of explaining our approach and the comparison with q-FFL. We truly appreciate your valuable feedback which helps us highlight our contributions and improve the clarity of the comparison with related methods.

---

> ### Author Response · Authors · 2021-09-02
> **Gentle reminder for response**
>
> We would like to gently remind the reviewer of any follow-up clarifications or questions that we can do our best to address in the remaining time. We hope our response adequately addressed your comments related to how fairness is achieved and a comparison with q-FFL and it has helped improve your opinion of our work. Please let us know if there are additional comments you have for us.

---

### Author Response · Authors · 2021-09-01
**Post-response period thank-you**

To all reviewers, we wish to thank you for spending the time reviewing our work and your valuable feedback. Furthermore, we truly appreciate your patience in going through our responses and raising the ratings. We will carefully address your comments in our revision.

---

### Decision · Program_Chairs · 2021-09-27

**Decision:**

Accept (Poster)

**Comment:**

This paper develops an interesting measure for user contribution in collaborative machine learning and an associated compensation mechanism. In general, the reviewers are positive about this paper.  The discussion has resolved several issues raised by the reviewers. Please update the paper to clarify the issues in the next revision.